# The pattern of nodal morphogen signaling is shaped by co-receptor expression

Nathan D Lord[1†‡*], Adam N Carte[1,2,3†], Philip B Abitua[1], Alexander F Schier[1,3,4*]

[1]Department of Molecular and Cellular Biology, Harvard University, Cambridge, United States; [2]Systems, Synthetic, and Quantitative Biology PhD Program, Harvard University, Cambridge, United States; [3]Biozentrum, University of Basel, Basel, Switzerland; [4]Allen Discovery Center for Cell Lineage Tracing, University of Washington, Seattle, United States

**\*For correspondence:**
ndlord@pitt.edu (NDL);
alex.schier@unibas.ch (AFS)

[†]These authors contributed equally to this work

**Present address:** [‡]Department of Computational and Systems Biology, University of Pittsburgh School of Medicine, Pittsburgh, United States

**Competing interests:** The authors declare that no competing interests exist.

**Abstract** Embryos must communicate instructions to their constituent cells over long distances. These instructions are often encoded in the concentration of signals called morphogens. In the textbook view, morphogen molecules diffuse from a localized source to form a concentration gradient, and target cells adopt fates by measuring the local morphogen concentration. However, natural patterning systems often incorporate numerous co-factors and extensive signaling feedback, suggesting that embryos require additional mechanisms to generate signaling patterns. Here, we examine the mechanisms of signaling pattern formation for the mesendoderm inducer Nodal during zebrafish embryogenesis. We find that Nodal signaling activity spans a normal range in the absence of signaling feedback and relay, suggesting that diffusion is sufficient for Nodal gradient formation. We further show that the range of endogenous Nodal ligands is set by the EGF-CFC co-receptor Oep: in the absence of Oep, Nodal activity spreads to form a nearly uniform distribution throughout the embryo. In turn, increasing Oep levels sensitizes cells to Nodal ligands. We recapitulate these experimental results with a computational model in which Oep regulates the diffusive spread of Nodal ligands by setting the rate of capture by target cells. This model predicts, and we confirm in vivo, the surprising observation that a failure to replenish Oep transforms the Nodal signaling gradient into a travelling wave. These results reveal that patterns of Nodal morphogen signaling are shaped by co-receptor-mediated restriction of ligand spread and sensitization of responding cells.

## Introduction

Developing embryos often transmit instructions using morphogens, diffusible signaling molecules that induce concentration-dependent responses in target cells. In the most common conception of morphogen function, ligands spread from a localized source to form a concentration gradient (*Lander, 2007*; *Müller et al., 2013*; *Stapornwongkul and Vincent, 2021*). Cells within the gradient infer their position by sensing the local ligand concentration and initiate a position-appropriate gene expression program (*Rogers and Schier, 2011*; *Stumpf, 1966*; *Wolpert, 1969*). Examples of gradient-driven patterning in animal embryos are plentiful; vertebrate germ layer induction (*McDowell and Gurdon, 1999*; *Schier, 2003*; *Shen, 2007*), dorsoventral organization of the neural tube (*Ericson et al., 1997*; *Yamada et al., 1993*), and digit patterning (*Raspopovic et al., 2014*; *Sheth et al., 2012*) all rely on graded profiles of signaling molecules. Principles derived from these examples have recently guided the design of synthetic patterning systems. Engineered gradients have been used to pattern fields of cultured human cells (*Li et al., 2018*; *Toda et al., 2020*) and to replace an endogenous morphogen gradient in the *Drosophila* wing disk (*Stapornwongkul et al., 2020*). The biological and physical processes that set the shape of morphogen gradients are

therefore of key importance to understanding developmental patterning and to the design of synthetic developmental systems.

Diffusion plays a central role in classical models of morphogen gradient formation (*Müller et al., 2013*; *Stapornwongkul and Vincent, 2021*). Ligand diffusion from a localized source is sufficient to create a concentration gradient that expands outward over time (*Berg, 1993*). Adding removal of the morphogen (through degradation, internalization, or other means) to the model confers stability (*Crick, 1970*). In such models, a steady-state gradient that does not further change in time can form (*Wartlick et al., 2009*). The shape of this steady-state gradient reflects a balance between ligand mobility and stability. Increasing the diffusion rate lengthens the gradient, whereas faster removal shortens it (*Wartlick et al., 2009*). Although simple, such diffusion-removal models approximate the behavior of several well-studied morphogens (*Kicheva et al., 2012*). Recent biophysical studies have shown that fluorescently tagged morphogens in *Drosophila* (*Kicheva et al., 2007*; *Zhou et al., 2012*) and zebrafish (*Müller et al., 2012*; *Yu et al., 2009*; *Zinski et al., 2017*) have diffusion rates consistent with known ranges of action. Similarly, receptor-mediated ligand capture provides a plausible mechanism for morphogen removal and has been shown to be a determinant of gradient range in some cases (*Yu et al., 2009*; *Baeg et al., 2004*; *Chen and Struhl, 1996*; *Lecuit and Cohen, 1998*; *Ribes and Briscoe, 2009*; *Scholpp and Brand, 2004*).

While these simple principles seem sufficient to explain gradient formation, diffusive transport may carry inherent limitations (*Müller et al., 2013*). For example, diffusing ligands could be difficult to contain without physical boundaries between tissues (*Kornberg and Guha, 2007*), and receptor saturation could preclude stable gradient formation (*Kerszberg and Wolpert, 1998*). Embryos may therefore need additional layers of control to spread signaling in a controlled fashion. Indeed, developmental signaling circuits often incorporate extensive feedback on morphogen production and sensing (*Rogers and Schier, 2011*; *Freeman, 2000*; *Freeman and Gurdon, 2002*; *Meinhardt, 2009*). In these systems, the shapes of signaling pattern can be determined by the action of feedback rather than the biophysical properties of signaling molecules. For example, it has been argued that positive feedback on ligand production can substitute for diffusion as a mechanism of morphogen dispersal. In this scheme, a cascade of short-range interactions—one tier of cells induces signal production in the next—can propagate signaling in space, even when the ligand itself is poorly diffusive. Such 'relay' mechanisms have been invoked to explain germ layer patterning in zebrafish (*van Boxtel et al., 2015*), as well as Wnt and Nodal signal spread in micropatterned stem cell colonies (*Chhabra et al., 2019*; *Liu, 2021*). Negative feedback can also shape signaling gradients, for example, by scaling patterns to fit tissue size (*Almuedo-Castillo et al., 2018*; *Ben-Zvi et al., 2008*), restricting signaling in space (*Chen and Struhl, 1996*), or turning off pathway activity when it is no longer needed (*van Boxtel et al., 2015*; *Golembo et al., 1996*). Due to the abundance of mechanisms that can contribute to signaling pattern shape, the mechanisms of gradient formation remain points of contention, even for well-studied morphogens.

Here, we examine the mechanism of gradient formation for the canonical morphogen Nodal. Nodals are TGFβ family ligands that function by binding to cell surface receptor complexes consisting of Type I and Type II activin receptors and EGF-CFC family co-receptors (*Schier, 2003*; *Shen, 2007*; *Gritsman et al., 1999*). Receptor complex formation induces phosphorylation and nuclear accumulation of the transcription factor Smad2, which cooperates with nuclear cofactors to activate Nodal target genes (*Massagué et al., 2005*). In early vertebrate embryos, Nodal signaling orchestrates germ layer patterning: exposure to high, intermediate, and low levels of Nodal correlates with selection of endodermal, mesodermal, and ectodermal fates, respectively (*Dougan et al., 2003*; *Gritsman et al., 2000*; *Thisse et al., 2000*; *Vincent et al., 2003*). Nodal signaling is under both positive and negative feedback control. Nodal ligands induce the expression of *nodal* genes (*Meno et al., 1999*), as well as of *leftys* (*Meno et al., 1999*; *Chen and Schier, 2002*), diffusible inhibitors of Nodal signaling. These feedback loops are conserved throughout vertebrates and therefore appear crucial to the function of the patterning circuit (*Shen, 2007*).

Zebrafish mesendoderm is patterned by two Nodal signals, Cyclops and Squint (*Shen, 2007*; *Dougan et al., 2003*). The physiologically relevant ligands are heterodimers between Cyclops or Squint and a third TGFβ family member, Vg1 (*Bisgrove et al., 2017*; *Montague and Schier, 2017*; *Pelliccia et al., 2017*). Gradient formation is initiated by secretion of Nodal ligands from the extraembryonic yolk syncytial layer (YSL), below the embryonic margin. Over time, the Nodal patterning circuit generates a gradient of signaling activity that, at the onset of gastrulation, extends

approximately 6–8 cell tiers from the margin (*van Boxtel et al., 2015*; *Dubrulle et al., 2015*; *Harvey and Smith, 2009*; *Rogers et al., 2017*). Mutations that markedly expand signaling range (e.g. *lefty1;lefty2*) result in profound phenotypic defects and embryonic lethality (*Rogers et al., 2017*). Proper development therefore relies on the generation of a correct Nodal signaling gradient.

Early studies with ectopically expressed Nodal ligands in zebrafish supported a model of diffusive spread (*Chen and Schier, 2001*). Direct observation of diffusion using GFP-tagged Cyclops and Squint ligands suggested short and intermediate ranges of activity, respectively (*Müller et al., 2012*). In this model, the distance that ligands can diffusively travel during the ~2 hr prior to gastrulation is a crucial determinant of gradient range. More recently, it was argued that Nodal signal spread was driven instead by positive feedback (*van Boxtel et al., 2015*). In this model, a feedback-driven relay spreads signaling activity away from the margin, and spread is stopped by the onset of Lefty production. In contrast to the diffusion-driven model, the range of signaling is set by the properties of the feedback circuit (e.g. the time required for a cell to switch on Nodal production and the delay in onset of Lefty production).

In this study, we re-examine the mechanisms that regulate Nodal signaling gradient formation in zebrafish embryos. We find that endogenous Nodal ligands can spread over a normal range in the absence of signaling feedback and relay, suggesting that diffusion is sufficient for gradient formation. Unexpectedly, we discover that the EGF-CFC co-receptor Oep is a potent regulator of the range of both Cyclops and Squint; in mutants lacking *oep*, Nodal activity is near-uniform throughout the embryo. We also find that Oep, although traditionally regarded as a permissive signaling factor, sets cell sensitivity to Nodal ligands. We incorporate these observations into a mathematical model for Nodal signal spread and predict that replenishment of Oep by zygotic expression is required for gradient stability. Finally, we verify a surprising prediction of the model: in zygotic *oep* mutants, which cannot replace Oep after it has been consumed, Nodal signaling propagates outward from the margin as a traveling wave. These findings illustrate how the embryo uses an unappreciated property of Oep—regulation of the rate of ligand capture—to set the range, shape, and intensity of the Nodal signaling gradient.

## Results

### The Nodal signaling gradient forms in the absence of feedback

The Nodal signaling gradient may reflect the diffusive properties of Nodal ligands secreted from the YSL or the action of signaling feedback and relay. To characterize the contribution of diffusion specifically, we set out to visualize the Nodal gradient in mutants that lack signaling feedback and relay altogether. This goal presented two key challenges. First, endogenous Nodal ligands have not been successfully visualized by antibody staining or fluorescent tagging in zebrafish. Second, knocking out the full complement of all known Nodal feedback regulators—for example *lefty1, lefty2, cyclops, squint, dpr2* (*Zhang et al., 2004*), etc—in combination is impractical. To address these two limitations, we were inspired by previous approaches for clone-mediated perturbations to morphogen gradients (*Baeg et al., 2004*; *Belenkaya et al., 2004*; *Cadigan et al., 1998*; *Eldar and Barkai, 2005*; *Entchev et al., 2000*) and developed a 'sensor' cell assay (*Figure 1A*). In this approach, we transplant Nodal-sensitive ('sensor') cells from a *gfp*-injected donor embryo to the margin of a host that is Nodal-insensitive and therefore lacks feedback. We then visualize signaling in the sensor cells by immunostaining for phosphorylated Smad2 (pSmad2) and GFP. Because host cells cannot respond to Nodal, they cannot modulate signal spread by either positive or negative feedback. For example, a transcriptional relay that spreads *nodal* expression would not form in this scenario. In addition, the sensor cells 'report' on their local Nodal concentration via pSmad2 staining intensity, enabling us to sample the activity of endogenous, untagged ligands. For the experiments described here, we use sensor cells from M*vg1* donors. These cells are Nodal-sensitive but cannot produce functional Nodal-Vg1 heterodimers and therefore cannot spread signaling via positive feedback (*Montague and Schier, 2017*). To pilot the sensor cell assay, we transplanted cells from an M*vg1* donor into a wild-type host (*Figure 1B*, upper panel). The M*vg1* sensors exhibited α-pSmad2 staining intensity similar to their wild-type neighbors, and quantification of staining across replicate embryos revealed similar signaling gradients for host and sensor cells (*Figure 1B*, lower panel; blue and red points, respectively). This result demonstrates that transplanted sensor cells accurately

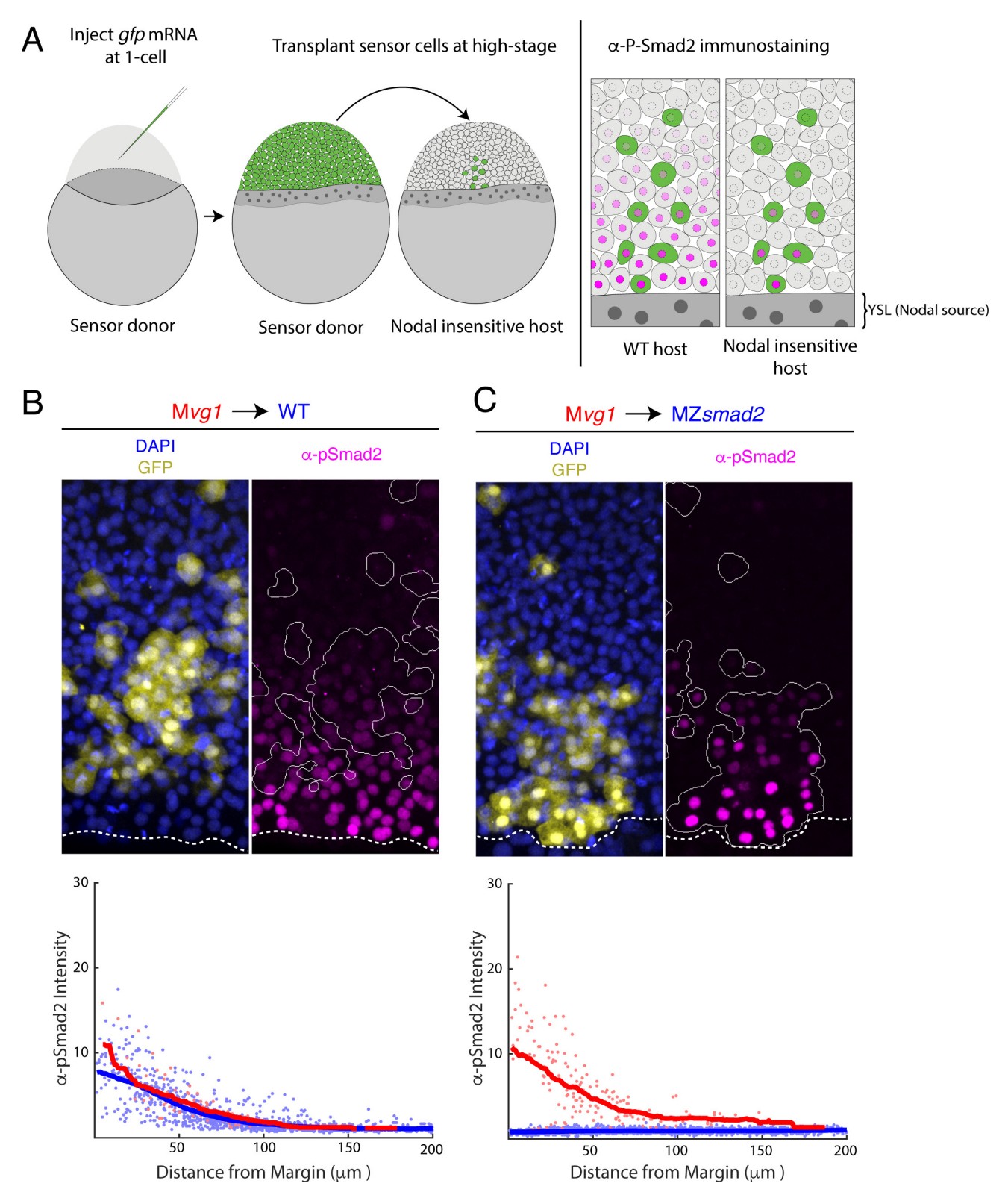

**Figure 1.** Nodal gradient formation in the absence of feedback. (**A**) Schematic of sensor cell assay. M*vg1* donor embryos were marked by injecting *gfp* mRNA at the 1-cell stage. At high stage, just before the onset of Nodal signaling, GFP-marked sensor cells were transplanted from the animal pole of the donor to the margin of a Nodal-insensitive host. At 50% epiboly, embryos were fixed and immunostained for GFP and Nodal signaling activity (α-pSmad2). Imaging of chimeric embryos (far right) enables inference of the gradient shape from α-pSmad2 staining (magenta) in sensor cells (green).
*Figure 1 continued on next page*

*Figure 1 continued*

Because host embryos lack the ability to respond to Nodal, YSL-derived Nodal ligands are responsible for the shape of the Nodal signaling gradient. (B) Control visualization of the Nodal signaling gradient in wild-type hosts using a sensor cell assay. Upper panel; M*vg1* sensor cells (yellow) were transplanted to the margin of a wild-type host. Nodal signaling was visualized by α-pSmad2 staining (magenta), and sensor cell boundaries were segmented with an automated pipeline (white curves). YSL boundaries are marked with dashed white curves. Lower panel; quantification of staining intensity in host (blue) and sensor (red) cells across replicate embryos. Nuclei were segmented from DAPI signal using an automated analysis pipeline implemented in MATLAB. Sensor and host cells were identified as being clearly GFP positive or negative, respectively. Solid curves represent sliding window averages. Plot was derived from three replicate embryos. (C) Sensor cell assay in MZ*smad2* host embryos. Upper panel; GFP-marked M*vg1* sensor cells (yellow) were transplanted to the margin of MZ*smad2* host embryos. Nodal signaling was visualized with α-pSmad2 staining (magenta). Sensor cell boundaries are marked with white outlines, and YSL boundaries are marked with dashed white curves. Lower panel; quantification of host (blue) and sensor (red) cell staining intensities were carried out as in (B). Plot was derived from six replicate embryos.

The online version of this article includes the following source data and figure supplement(s) for figure 1:

**Source data 1.** In *Figure 1B*, sensor cell assay results were quantified by segmenting nuclei and classifying each nucleus as host- or donor-derived by GFP intensity.

**Source data 2.** In *Figure 1C*, sensor cell assay results were quantified by segmenting nuclei and classifying each nucleus as host- or donor-derived by GFP intensity.

**Figure supplement 1.** MZ*smad2*, M*vg1*, and MZ*oep* mutants lack pSmad2.

**Figure supplement 2.** MZ*smad2* and MZ*oep* embryos have intact Nodal sources.

report on their local signaling environment. We further note that sensor cell migration after transplant does not appear to compromise the assay, as sensor cells exhibit signaling intensities appropriate for their position at the time of embryo fixation. This outcome is consistent with previous observations that cell rearrangement at the margin is minimal prior to gastrulation (*Dubrulle et al., 2015*; *Helde et al., 1994*; *Wilson et al., 1993*).

We next applied this approach to MZ*smad2* host embryos, which lack all Nodal signaling. Smad2 is required to activate Nodal-dependent gene expression, and zebrafish MZ*smad2* embryos phenocopy mutants lacking Nodal ligands (*Dubrulle et al., 2015*). We verified that MZ*smad2* embryos lack pSmad2 (*Figure 1—figure supplement 1*) but continue to express *cyclops* and *squint* in the YSL (*Figure 1—figure supplement 2*). Expression of both Nodals was excluded from the blastoderm, confirming that these mutants are incapable of Nodal autoregulation (*Figure 1—figure supplement 2*). M*vg1* sensor cells transplanted into MZ*smad2* mutants exhibit clear Nodal signaling activity several cell tiers from the margin (*Figure 1C*, upper panel), while signaling was completely absent in host cells. Quantification of staining in MZ*smad2* hosts (*Figure 1C*, lower panel) revealed a Nodal signaling gradient similar in range to that of wild-type controls (*Figure 1B.*, lower panel; half-distances of 45 and 37 μm for MZ*smad2* and wild type, respectively). Together, these experiments suggest that YSL-derived Nodal ligands can form a gradient of normal range without help from signaling feedback and relay.

## Nodal signaling range is expanded in the absence of Oep

The above results support a model in which diffusion drives Nodal spread. However, it remains unclear how the embryo sets the range of ligand dispersal. Biophysical studies with GFP-tagged Nodals suggest that ligand mobility may be hindered by interaction with extracellular factors, as measured diffusion rates for both Cyclops and Squint are >10 fold lower than for free GFP (*Müller et al., 2012*). However, no factors that explain hindered mobility of endogenous ligands have been identified. Cell surface receptor complexes are clear candidates for this role (*Wang et al., 2016*), because transient ligand capture or receptor-mediated endocytosis could constrain the gradient (*Wartlick et al., 2009*), and receptors have been shown to regulate gradient range for other signals (*Baeg et al., 2004*; *Chen and Struhl, 1996*; *Lecuit and Cohen, 1998*; *Okabe et al., 2014*).

To test whether receptor complex components regulate the range of Nodal signaling, we performed sensor cell transplants in embryos lacking the essential Nodal co-receptor Oep (MZ*oep* mutants *Gritsman et al., 1999*). We found that M*vg1* sensor cells detected Nodal activity over a dramatically longer range in MZ*oep* hosts than in wild-type controls (*Figure 2A,B*). Indeed, transplanting sensor cells to the animal pole revealed that Nodal ligand activity can be detected throughout the embryo when Oep is absent (*Figure 2D,E*). To test whether loss of Oep affects both Nodal ligands similarly, we performed sensor cell assays in MZ*oep;sqt* and MZ*oep;cyc* double mutants.

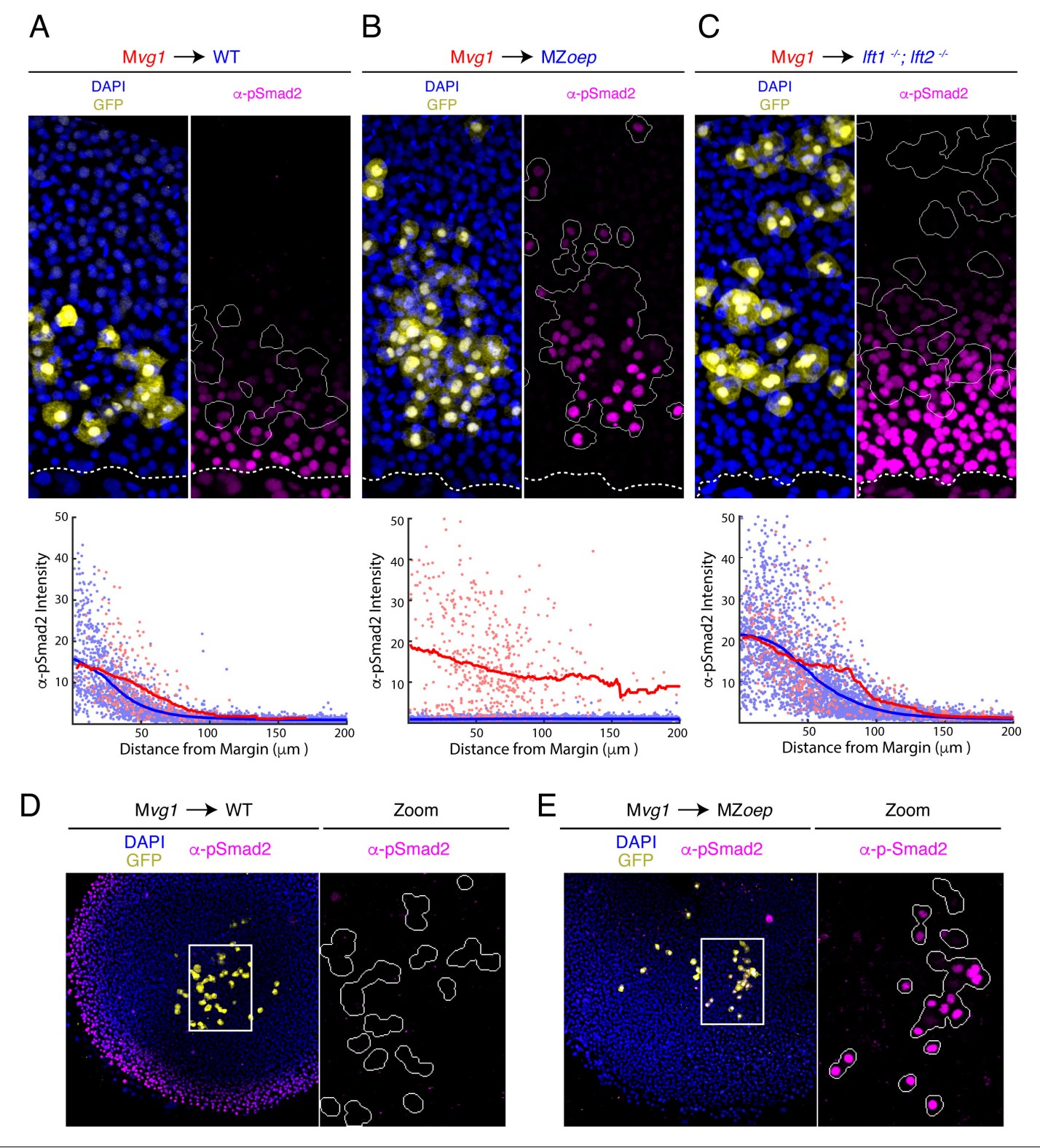

**Figure 2.** The Nodal gradient is expanded in MZ*oep* mutants. (A-C) Sensor cell assay and gradient quantifications in (A) wild type, (B) MZ*oep*, and (C) *lft1⁻/⁻;lft2⁻/⁻* embryos. M*vg1* sensor cells were marked with GFP (yellow) and transplanted to the margin of host embryos. Nodal signaling activity is measured by α-pSmad2 immunostaining (magenta). YSL boundaries are marked with dashed curves and sensor cell boundaries are outlined in solid white in all α-pSmad2 panels. Gradient quantifications for each experiment are below images; host and sensor cell staining intensities are plotted as blue and red points, respectively. Sliding window averages are plotted as solid curves. Plots for wild type, MZ*oep*, and *lft1⁻/⁻;lft2⁻/⁻* backgrounds were

*Figure 2 continued on next page*

*Figure 2 continued*

derived from 8, 10, and 8 replicate embryos, respectively. Decay parameters for single-exponential model fits (±95% confidence bounds) are $-0.02 \pm 0.004$ µm$^{-1}$, $-0.007 \pm 0.002$ µm$^{-1}$ and $-0.013 \pm 0.002$ µm$^{-1}$ for wild-type, MZ*oep* and *lft1*$^{-/-}$;*lft2*$^{-/-}$ hosts, respectively. (D) Left panel; M*vg1* sensor cells (yellow) were transplanted directly to the animal pole of a wild-type host. The endogenous Nodal signaling gradient is visible at the embryonic margin (magenta). White box highlights region expanded for detail view in right panel. Right panel; Nodal signaling activity is absent in both host and sensor cells. (E) Left panel; M*vg1* sensor cells (yellow) were transplanted to the animal pole of an MZ*oep* embryo. Nodal signaling is absent at the embryonic margin. White box highlights region expanded in the right panel. Right; sensor cells detect Nodal at the animal pole (magenta).

The online version of this article includes the following source data and figure supplement(s) for figure 2:

**Source data 1.** In *Figure 2A*, sensor cell assay results were quantified by segmenting nuclei and classifying each nucleus as host- or donor-derived by GFP intensity.

**Source data 2.** In *Figure 2B*, sensor cell assay results were quantified by segmenting nuclei and classifying each nucleus as host- or donor-derived by GFP intensity.

**Source data 3.** In *Figure 2C*, sensor cell assay results were quantified by segmenting nuclei and classifying each nucleus as host- or donor-derived by GFP intensity.

**Figure supplement 1.** Cyclops and Squint signal over a long range in the absence of Oep.

**Figure supplement 2.** Clustering does not contribute to Nodal sensitivity in sensor cells.

**Figure supplement 3.** The Nodal ligand gradient is shaped by *oep* expression.

Loss of Oep led to an expanded range of action for both Cyclops (i.e. in MZ*oep;sqt* mutants) and Squint (i.e. in MZ*oep;cyc* mutants), and the signaling ranges in both double mutants were comparable to that observed in the MZ*oep* single mutant (*Figure 2—figure supplement 1*). We note that long-range Nodal signaling in *oep* mutants does not reflect residual Nodal signaling between M*vg1* sensor cells, as signaling intensity was independent of sensor cell density (*Figure 2—figure supplement 2*). Although endogenous Nodal ligands have not been detectable to date and the sensor assay is the most sensitive reporter for signaling by Nodal ligands, we ectopically expressed GFP-tagged Squint in a transplanted clone of source cells. Direct ligand visualization also revealed an expanded range of secreted Nodal in MZ*oep* mutants compared to wild type (*Figure 2—figure supplement 3*).

In summary, the sensor assays reveal a remarkable gradient expansion in MZ*oep* mutants when compared with the effect of other mutations that alter Nodal signaling range. For example, the expansion of the signaling gradient in *lefty1;lefty2* mutant embryos, which lack negative feedback on Nodal signaling (*Rogers et al., 2017*; *Figure 2C*), is mild compared to our observations in MZ*oep* embryos (*Figure 2B,C*). These results demonstrate that receptor complexes play key roles in constraining the spread of Nodal signals from the YSL.

## Oep regulates the range and intensity of Nodal signaling through ligand capture

EGF-CFC proteins such as Oep are typically regarded as permissive factors for Nodal signaling. Oep facilitates the assembly of receptor-ligand complexes but is not thought to regulate signaling beyond conferring competence (*Zhang et al., 1998*). However, our finding that Nodal ligand range is expanded in the absence of Oep suggests that it has unappreciated regulatory roles. The simplest way to accommodate this result is to stipulate that Oep levels set the rate of capture of diffusing Nodal ligands. Through this mechanism, Oep could control the range of Nodal activity by regulating the rate of receptor-mediated ligand internalization (i.e. the effective ligand degradation rate). This model makes two testable predictions. First, increasing Oep levels should enhance cell sensitivity to Nodal ligands by facilitating capture by receptor complexes. Second, increasing Oep levels should reduce the range of Nodal signaling by increasing the effective degradation rate.

To test whether Oep regulates cell sensitivity, we asked whether overexpressing *oep* in sensor cells increases their responsiveness to endogenous Nodals. We transplanted cells from M*vg1* embryos injected with *oep* and *gfp* mRNAs or with *gfp* alone to the margin of wild-type embryos and immunostained for GFP and pSmad2. Sensors with increased Oep levels stained more brightly for pSmad2 than neighboring host cells (*Figure 3B*), while sensors injected with *gfp* alone matched the behavior of their neighbors (*Figure 3A*). Interestingly, we found that the *oep*-overexpressing sensors detected Nodal further from the margin than the host cells, suggesting that the Nodal ligand gradient extends beyond the domain of detectable signaling in normal embryos (*Figure 3B*).

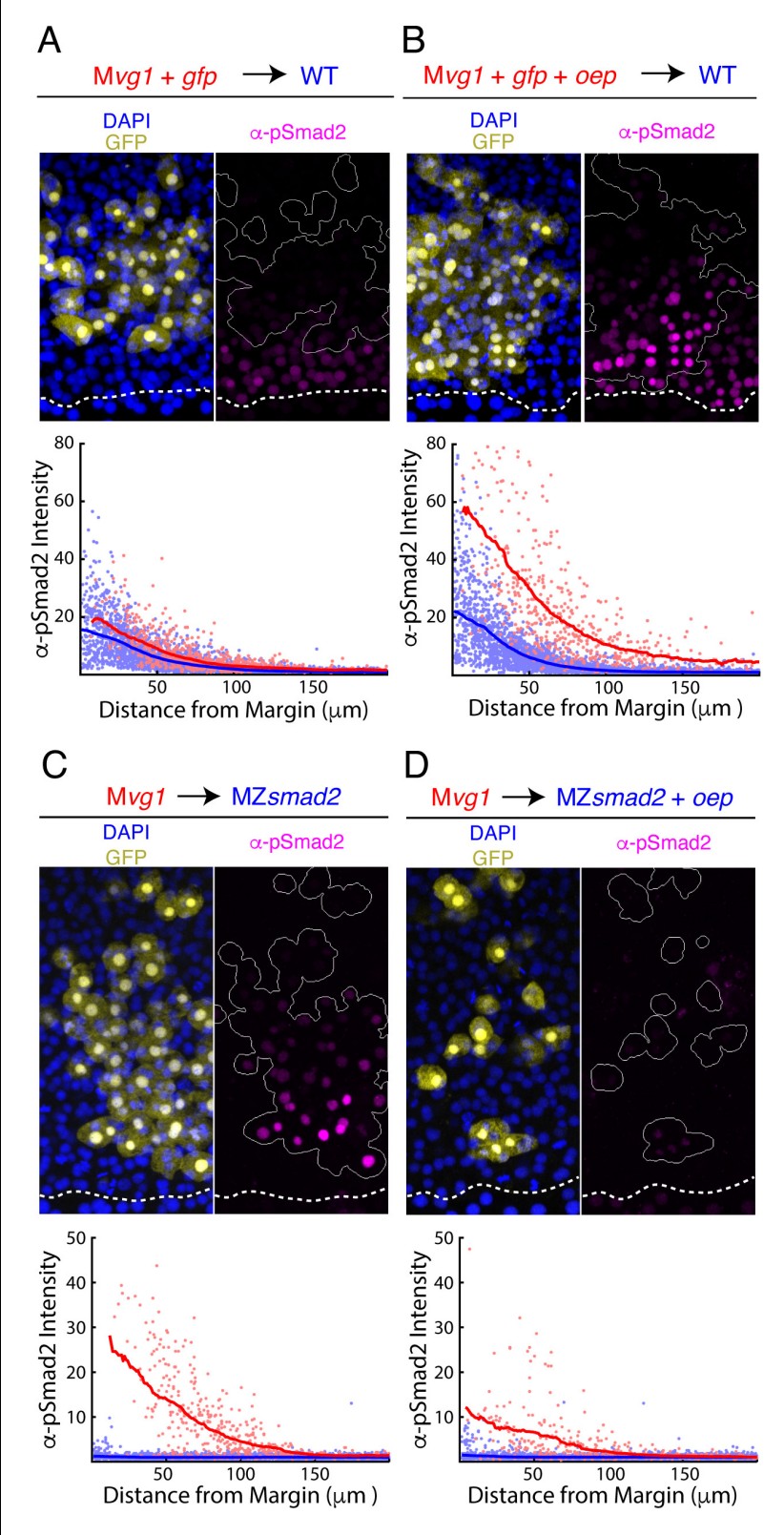

**Figure 3.** Oep levels regulate Nodal ligand capture and signaling range. (**A–B**) Oep overexpression increases sensitivity to Nodal ligands. (**A**) Upper panel: control transplant of GFP-marked M*vg1* sensor cells (yellow) to the margin of wild-type hosts. Nodal signaling activity was measured by α-pSmad2 immunostaining (magenta). In all panels, YSL boundaries are marked with dashed white curves, and sensor cells have been outlined in solid white in *Figure 3 continued on next page*

*Figure 3 continued*

all α-pSmad2 panels. Lower panel: quantification of Nodal signaling in sensor (red) and host cells (blue) across replicate embryos. Sliding window averages are plotted as solid curves. Plot was derived from eight replicate embryos. (B) Upper panel: transplant of sensor cells from an M*vg1* donor injected with *gfp* and 110 pg *oep* mRNA at the one-cell stage to the margin of wild-type hosts. Sensor cells (yellow) exhibit enhanced Nodal signaling activity (magenta) compared to their host-derived neighbors. Lower panel; staining of host (blue) and sensor (red) cells was quantified as in (A). Plot was derived from nine replicate embryos. (C-D) Oep overexpression restricts Nodal spread. (C) Upper panel: sensor cell measurement of the Nodal gradient in MZ*smad2* embryos. M*vg1* sensor cells were marked with GFP (yellow), and Nodal signaling activity was measured by α-pSmad2 immunostaining (magenta). Lower panel: quantification of Nodal signaling in sensor (red) and host cells (blue) was quantified as in (A). Plot was derived from nine replicate embryos. (D) Upper panel: M*vg1* sensor cell measurement of the Nodal signaling gradient in MZ*smad2* hosts injected with 110 pg *oep* mRNA at the one-cell stage. Lower panel; gradients were quantified as in (A). Plot was derived from nine replicate embryos.

The online version of this article includes the following source data and figure supplement(s) for figure 3:

**Source data 1.** In *Figure 3A*, sensor cell assay results were quantified by segmenting nuclei and classifying each nucleus as host- or donor-derived by GFP intensity.
**Source data 2.** In *Figure 3B*, sensor cell assay results were quantified by segmenting nuclei and classifying each nucleus as host- or donor-derived by GFP intensity.
**Source data 3.** In *Figure 3C*, sensor cell assay results were quantified by segmenting nuclei and classifying each nucleus as host- or donor-derived by GFP intensity.
**Source data 4.** In *Figure 3D*, sensor cell assay results were quantified by segmenting nuclei and classifying each nucleus as host- or donor-derived by GFP intensity.
**Figure supplement 1.** Nodal ligand range is expanded in MZ*oep* mutants.

We note that the increased sensitivity of the *oep*-overexpressing sensors does not reflect the action of hyperactive-positive feedback on Nodal production, as M*vg1* cells are incapable of producing functional Nodal-Vg1 heterodimers. These results suggest that, in addition to being required for signaling competence, Oep regulates sensitivity to Nodal ligands.

To test whether Oep levels modulate Nodal range, we asked whether overexpression of *oep* could restrict signaling. We performed sensor cell assays in MZ*smad2* hosts injected with *oep* mRNA at the one-cell stage. Overexpression of Oep indeed reduced the range and intensity of Nodal signaling (*Figure 3D*) when compared with uninjected hosts (*Figure 3C*). We note that the choice of MZ*smad2* hosts was important for interpretation of the experiment. As Oep sensitizes cells to Nodal ligands, increasing expression in signaling-competent host embryos could lead to increased signaling by triggering Nodal positive feedback. Nodal signaling is disabled downstream of the receptor in MZ*smad2* mutants, allowing us to specifically test Oep's role in regulating ligand range without this confound.

To further test the idea that Oep restricts Nodal spread, we analyzed the distribution of fluorescently-tagged Squint in embryos expressing excess *oep*. In the first experiment, we visualized the range of Squint-sfGFP gradients generated by transplanted source cells in hosts lacking *oep* (MZ*oep*) and hosts overexpressing *oep* (MZ*oep* injected with *oep* mRNA). Consistent with our sensor cell results, overexpression of *oep* resulted in marked shortening of the Squint-sfGFP gradient relative to MZ*oep* (*Figure 2—figure supplement 3*). In a second experiment, we expressed Halo-tagged Vg1 and Squint in the YSL and monitored their accumulation in sensor cells (*Figure 3—figure supplement 1*). Embryos producing tagged ligands were generated by injecting mRNAs encoding *vg1-halotag* and *squint* directly into the YSL shortly after its formation (1k-cell stage). To concentrate and clearly visualize the Halo-tagged ligand, we transplanted sensor cells from a donor embryo injected with *oep* mRNA to the animal pole, akin to a morphotrap approach (*Stapornwongkul et al., 2020*; *Almuedo-Castillo et al., 2018*; *Harmansa et al., 2017*; *Harmansa et al., 2015*). Halo-tagged ligand accumulated in the animal pole sensors in MZ*oep* hosts but not in wild-type hosts. This accumulation was prevented by overexpressing *oep* in the MZ*oep* hosts. Together, these results indicate that Oep regulates both the range and intensity of Nodal signaling.

## A simple model incorporating Oep-Nodal interaction reproduces experimental observations

We formulated a simple mathematical model of Nodal gradient formation to explore whether Oep-mediated capture of diffusing Nodal ligands is sufficient to explain our experimental data (*Figure 4A*). In the model, Nodal is secreted at a constant rate at one end of a two-dimensional tissue and diffuses freely until it is captured by a free receptor complex. We stipulate that ligand-receptor association follows pseudo first-order kinetics (i.e. that the free receptor concentration can be regarded as constant) and that internalization of receptor-ligand complexes is also first-order. To track integration of signaling activity, we also incorporate phosphorylation of Smad2 with a rate proportional to ligand-receptor complex concentration. Where possible, parameter values were taken from the literature. Model details and a summary of the rates used in simulations are presented in *Supplementary file 1*.

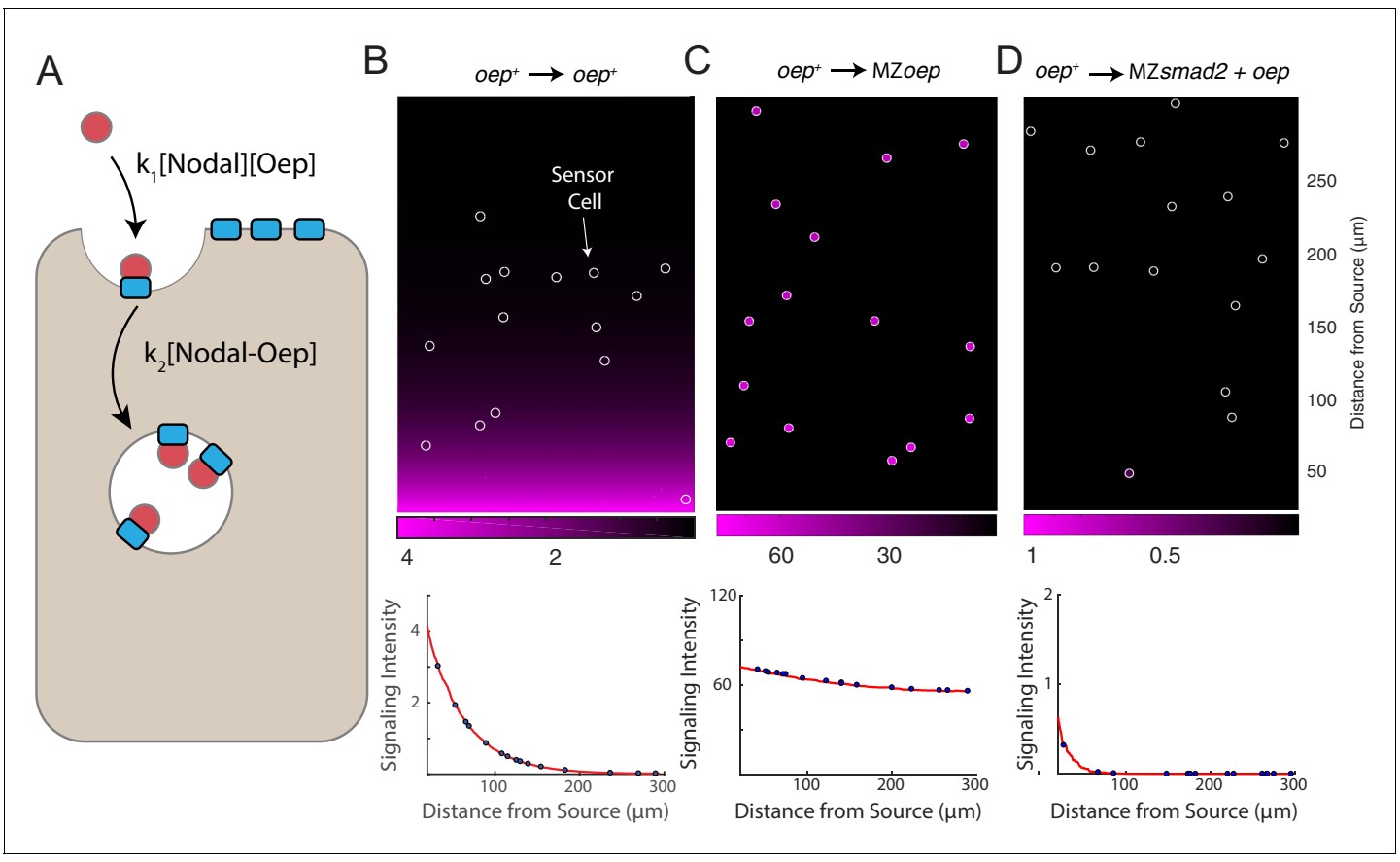

**Figure 4.** A simple model of Nodal diffusion and capture reproduces experimental observations. (**A**) Schematic of Nodal diffusion-capture model. Simulations were performed on a two-dimensional tissue of 100 μm x 300 μm. Nodal molecules are secreted at a constant rate from a localized source at one boundary of the tissue (i.e. 0 < x < 5 μm) and diffuse freely until capture by cell surface receptors ('Oep'). Ligand-receptor complexes are removed from the system by internalization. To track signaling activity, Smad2 phosphorylation is simulated with rate proportional to the concentration of receptor-ligand complexes. (**B-D**) Simulation of transplant experiments. In each simulation, the behavior of sensor cells (white outlines) is compared with the behavior of the host embryo (remainder of tissue). Parameters were independently set for host and sensor regions, allowing for simulation of experiments with mutations and overexpression. Signaling activity (i.e. [pSmad2]) is plotted in magenta. Upper panels present representative simulations with randomly-positioned sensor cells. Lower panels depict quantified signaling intensities for sensor cells from the panel above (blue points) and average intensities derived from replicate simulations (red curves). (**B**) Wild-type gradient simulation. Sensor cells with normal Oep levels are transplanted into a host with normal Oep levels. A stable gradient forms, and signaling is identical in sensor cells and neighboring regions. (**C**) Gradient expansion in MZ*oep* mutants. Sensor cells contain normal Oep levels, but host cells lack Oep. Sensor cells detect ligand throughout the tissue. (**D**) Gradient contraction with *oep* overexpression. Sensor cells contain normal Oep levels, whereas host cells lack Smad2, but overexpress *oep*. Signaling is absent in the host tissue—due to lack of Smad2—but elevated receptor expression restricts Nodal spread to the sensors.

This simple model reproduces a signaling gradient with a scale and shape consistent with our observations in wild-type embryos (*Figure 4B*). To reproduce our experimental data, we simulated sensor cell assays (*Figure 4B–D*, sensor cells highlighted with white outlines). Expansion of the Nodal ligand gradient in MZ*oep* mutants can be reproduced by simulating 'hosts' with the receptor concentration set to zero (*Figure 4C*). Similarly, restriction of signaling range via *oep* overexpression could be reproduced by increasing receptor levels in host cells, but not in the sensors (*Figure 4D*). A model in which Nodal capture rate is set by Oep concentration can therefore reproduce our major experimental findings.

## Loss of Oep replenishment transforms nodal signaling dynamics

The simplified model presented above assumes that free receptor cannot be depleted by ligand binding. While convenient, this condition may be difficult for the embryo to achieve in practice. For example, maintaining receptors at high concentration would preclude depletion but could also prevent ligand from traveling long distances before capture. Another way for the embryo to avoid depletion would be to continually replace receptor components as they are consumed by ligand binding. To explore the role of receptor complex replacement in gradient formation, we explicitly incorporated receptor production and degradation into the model (*Figure 5A*).

Simulations incorporating receptor production and consumption generate stable exponential gradients (*Figure 5B*) with length scales comparable to our measurements in zebrafish embryos. To test the consequences of losing co-receptor replacement, we simulated gradient formation in a system that begins with a finite supply of free receptors that are not replaced. This change results in a surprising transformation of Nodal signaling dynamics; simulations with finite co-receptor supply generate a traveling wave of Nodal signaling that propagates outward from the ligand source (*Figure 5C*, magenta). These dynamics reflect the gradual consumption of co-receptors due to ligand binding and subsequent endocytosis (*Figure 5C*, cyan). Initially, when co-receptor is plentiful, the source generates a decaying gradient of signaling. Over time, receptors close to the source are depleted, allowing Nodal ligands to rapidly traverse this space, ultimately reaching a new population of sensitive cells. We note that wave formation does not critically depend on our assumptions regarding the mechanism of co-receptor downregulation; a model that incorporates Oep trafficking and recycling also supports our key conclusions (*Figure 5—figure supplement 1*). In sum, these simulations raise the possibility that co-receptor replenishment is a key determinant of the Nodal gradient shape.

To test this idea, we measured Nodal signaling patterns in zygotic *oep* mutants (Z*oep*) (*Schier et al., 1997*). This background reproduces the key assumptions of the model above: Z*oep* mutants begin with a finite supply of maternally provided *oep* mRNA but cannot express additional *oep* from the zygotic genome (*Zhang et al., 1998*). Indeed, previous studies have shown that maternally deposited *oep* mRNA is undetectable in Z*oep* mutants by germ ring stage (*Zhang et al., 1998*) and that *oep* mRNA is depleted from wild-type embryos by 60% epiboly in the absence of zygotic transcription (*Vopalensky et al., 2018*). We performed α-pSmad2 immunostaining in wild-type and Z*oep* mutant embryos at three timepoints following the initiation of Nodal secretion (dome, 50% epiboly and shield stages). Consistent with previous observations, the wild-type Nodal signaling profile monotonically decreases from the margin, decaying to background over ~8 cell tiers (*Figure 5D*). Strikingly, in Z*oep* mutants, Nodal signaling is restricted to the margin at dome stage (*Figure 5E*, left), but propagates outward to form a broad band of signaling by shield stage (*Figure 5E*, right). As predicted by the model, loss of co-receptor replacement by zygotic expression thus transforms a steady-state exponential gradient into a wave of Nodal signaling that propagates toward the animal pole. We note that, in accordance with model simulations, overall signaling intensity is lower in Z*oep* mutants due to lower overall co-receptor levels (*Figure 5F*). These results highlight the importance of continued co-receptor replacement in shaping the pattern of Nodal signaling.

## Discussion

In this study, we set out to identify mechanisms that determine the Nodal signaling gradient range and shape. We find that endogenous Nodals secreted from the YSL can drive signaling over a normal range in the absence of feedback and relay mechanisms (*Figure 1*). We go on to demonstrate that expression of Oep, a Nodal co-receptor, regulates the spread (*Figure 2*), potency (*Figure 3*),

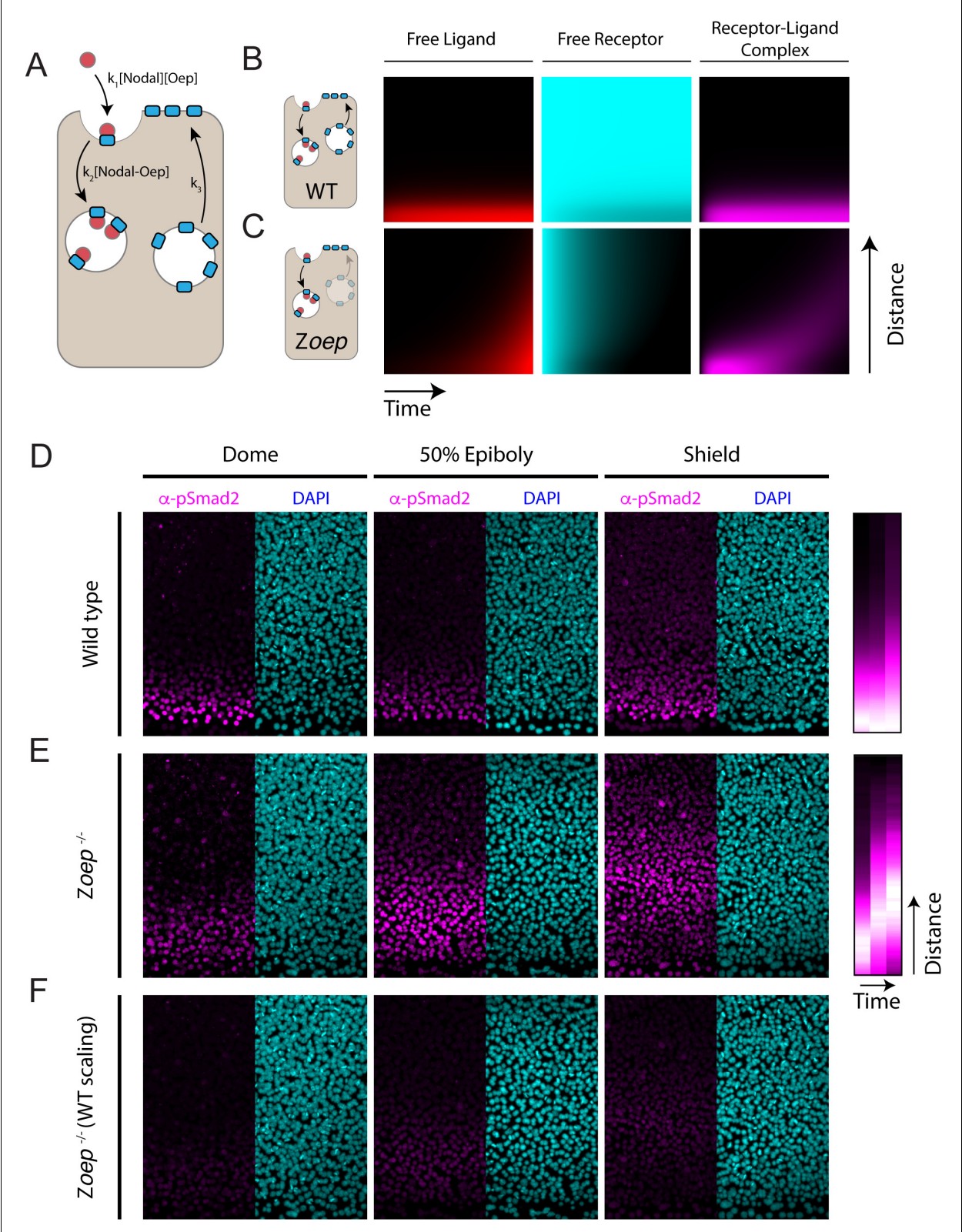

**Figure 5.** Loss of Oep replacement destabilizes the Nodal signaling gradient. (**A**) Schematic of model incorporating production and consumption of receptors. Simulations presented here were performed on a one-dimensional tissue with length 300 μm. Oep replacement is assumed to be constant with rate $k_3$, and Oep removal reflects a combination of constitutive and ligand-dependent endocytosis. In panels **A and B**, simulations are presented as kymographs; each image column shows the state of the system with the source at the bottom and animal pole at the top. Time proceeds from left to
*Figure 5 continued on next page*

*Figure 5 continued*

right. (**B**) Simulation of a wild-type gradient. With continual receptor replacement, the system achieves an exponential steady state gradient with length scale set by the ligand diffusion rate and receptor abundance. The free ligand, free receptor, and receptor-ligand complex concentrations are plotted from left to right in red, cyan, and magenta, respectively. (**C**) Simulation of gradient formation in a zygotic *oep* mutant. Simulation details are identical to (**B**), but with receptor replacement rate ($k_3$) set to zero. The system fails to establish a steady state due to gradual consumption and degradation of receptors. Over time, the Nodal ligand gradient expands (red) to drive a propagating wave of signaling activity (i.e. receptor occupancy, magenta). (**D**) Time course of Nodal signaling activity in wild-type embryos. Representative α-pSmad2 (magenta) and DAPI (cyan) are shown for dome, 50% epiboly and shield stages (left, middle and right panels, respectively). Compilation of signaling gradients across replicates (far right) shows the establishment of the signaling gradient. Composite gradients were derived from 5, 6, and 6 replicate embryos for dome, 50% epiboly and shield stages, respectively. (**E**) Time course of Nodal signaling activity in zygotic *oep* mutants. Over time, the signaling pattern evolves from a gradient (dome stage) to a band displaced far from the margin (shield) as the wave travels outward. Compilation of signaling gradients across replicates (far right) illustrates the outward propagation of signaling. Composite gradients were derived from 7, 6, and 3 replicate embryos for dome, 50% epiboly, and shield stages, respectively. (**F**) Time course of Nodal signaling activity in zygotic *oep* mutants presented with pixel scaling equal to that used in (**D**). In accord with simulations, the wave of signaling propagates with a lower intensity than signaling at the margin of wild-type embryos.

The online version of this article includes the following figure supplement(s) for figure 5:

**Figure supplement 1.** An endocytic trafficking model predicts wave formation in Z*oep* mutants.

and shape (*Figure 5*) of Nodal activity. We present a simple computational model that explains the Nodal signaling gradient in terms of free ligand diffusion and binding to cell surface receptor complexes (*Figure 4*). In this description, Oep regulates the range of ligand spread and sensitivity of embryonic cells by setting the rate of ligand capture. This simple model accommodates our main observations—gradient formation without feedback, increased signaling range in co-receptor mutants, restricted range with increased co-receptor expression, and a signaling wave in the absence of co-receptor replenishment.

Diffusion has long been regarded as an attractive mechanism for signal dispersal in tissues (*Crick, 1970*). Indeed, signaling patterns consistent with simple diffusion-degradation mechanisms— for example single-exponential gradients with length scales of ~10–100 μm— are common in developing tissues (*Kicheva et al., 2012*). Viewed in this light, the regulatory complexity of developmental patterning circuits is striking; if diffusion is sufficient to generate observed signaling patterns, why are co-factors and extensive feedback loops so common? One possible answer is that diffusion carries inherent disadvantages. For example, it has been argued that diffusible ligands would be impractical to contain without physical boundaries (*Kornberg and Guha, 2007*), and that diffusion-driven gradients would not be a reliable source of positional information (*Wolpert, 2016*). We and others have proposed feedback-centered Nodal patterning models that offer a way around these dilemmas (*Müller et al., 2012*; *van Boxtel et al., 2015*; *Chen and Schier, 2002*; *Rogers et al., 2017*; *Nakamura et al., 2006*). However, it has not been possible to clearly test whether feedback is required for the dispersal of endogenous ligands. This study is the first to examine the shape of the Nodal signaling gradient in the absence of feedback and relay. We found that a gradient of approximately normal range and shape can form even when feedback is disabled.

Recent studies in zebrafish embryos (*van Boxtel et al., 2015*) and human gastruloids (*Liu, 2021*) have proposed that long-range Nodal signaling relies on a positive feedback-driven relay. In zebrafish, this conclusion was based on the observations that Nodal signaling induces *nodal* gene expression (*Meno et al., 1999*) and the expression domain of a synthetic Nodal reporter gene coincides with, but does not extend beyond, the *nodal* expression domain (*van Boxtel et al., 2015*). While these findings are consistent with relay-driven transport, these previous zebrafish studies did not test whether the range of Nodal signaling indeed depends on *nodal* autoregulation and contracts when autoinduction is disrupted. Our findings directly address this question and reveal that relay mechanisms are not necessary for the generation of a Nodal signaling gradient in zebrafish. *Nodal* gene expression in the YSL is sufficient to establish a Nodal signaling gradient.

In human gastruloids, engineered gradients created with juxtaposed 'sender' and 'receiver' cells revealed that Nodal signaling is attenuated when receiver cells are *nodal* mutants (*Liu, 2021*). This experiment demonstrates that *nodal* autoregulation supports the spread of Nodal signaling; however, since *nodal* expression in sender cells was also reduced in this context, it remains unclear if autoregulation is required for maintaining the initial *nodal* source or for generating a relay of *nodal* expression.

It is conceivable that different tissues require distinct implementations of the Nodal-Lefty patterning system. For example, the rapid pace of zebrafish mesendodermal patterning may make diffusion the only viable Nodal transport mechanism, while slower development in mammalian embryos may permit the use of multi-step, feedback-driven transport mechanisms. Different features of the Nodal-Lefty system (activator-inhibitor signaling; differential diffusivity; positive and negative feedback) might be distinctly employed for pattern formation in different contexts.

Our study identifies new roles for EGF-CFC co-receptors in Nodal signaling. Oep has been traditionally regarded as a permissive factor for signaling *Zhang et al., 1998*; it facilitates Nodal association with Activin receptors (*Cheng et al., 2003*; *Reissmann et al., 2001*; *Yeo and Whitman, 2001*) but was not thought to regulate gradient shape or cell sensitivity (*Gritsman et al., 1999*; *Zhang et al., 1998*). Our observations suggest that— similar to receptors for Dpp (*Lecuit and Cohen, 1998*), Hh (*Chen and Struhl, 1996*) and Wg (*Baeg et al., 2004*)—Oep is a key determinant of the mobility and potency of its cognate ligand. Indeed, far from being a bystander in gradient formation, Oep is one of the strongest regulators of Nodal range yet discovered. This finding also suggests a potential explanation for a key feature of the Nodal patterning circuit: differential diffusivity between Nodal ligands and Lefty proteins. GFP-tagged Cyclops and Squint diffuse substantially slower than free GFP, whereas tagged Lefty proteins diffuse rapidly (*Müller et al., 2012*). This feature of Nodal ligands is consistent with a hindered diffusion model in which interactions with immobile binding partners leads to a slow 'effective' diffusion rate, even if free molecules diffuse rapidly (*Müller et al., 2013*; *Müller et al., 2012*). Our data raise the possibility that the differential diffusivity of Nodal and Lefty proteins originates in rates of capture by available receptor complexes.

Oep-mediated ligand capture and signaling sensitization results in short-range enhancement and long-range inhibition of Nodal signaling: close to the Nodal source, Oep binds Nodal and stimulates signaling, whereas far from the source, little Nodal is available due to Oep-mediated capture close to the source. Despite its distinct molecular roles, the Nodal signaling factor Oep thus has a function reminiscent of the Nodal inhibitor Lefty. Lefty is produced at the margin, but diffuses rapidly to inhibit Nodal signaling far from the source. A common theme for Nodal regulators is therefore to counteract the inherent potential for long-range Nodal diffusion and signaling and to restrict Nodal signaling to a domain near the ligand source.

Our results suggest that the embryo's strategy for replenishing Oep is a key point of control over the signaling pattern. We found that, without this replacement, the Nodal signaling pattern is qualitatively transformed from a stable gradient into a propagating wave. Interestingly, a signaling wave of this type was predicted in a theoretical study of morphogen gradient formation by *Kerszberg and Wolpert, 1998*. In fact, they used this phenomenon to argue that receptor saturation would make stable gradients difficult to achieve by diffusive transport. Our results suggest that consumption of receptors can create precisely this type of unstable behavior, but that the embryo achieves a stable gradient through continual turnover of the receptor pool. Though not employed during mesendodermal patterning, this phenomenon could provide a simple means of repurposing the Nodal patterning circuit to create dynamic waves of signaling in other contexts. For example, in left-right patterning the consumption of Oep by Nodal might support the anterior spread of the expression of the Nodal gene *southpaw* (*Long et al., 2003*). More generally, signaling waves have emerged as a mechanism to coordinate diverse processes such as cell migration (*Aoki et al., 2017*), tissue regeneration (*De Simone et al., 2021*), and apoptosis (*Cheng and Ferrell, 2018*). Signaling feedback is generally invoked to explain these phenomena. However, our results suggest receptor depletion as an alternative, feedback-free mechanism of signaling wave formation. Finally, we speculate that the precise dynamics of Oep replacement might contribute additional interesting functions to patterning systems. For example, signaling-dependent receptor expression could confer robustness to fluctuations in source-derived morphogen production (*Barkai and Shilo, 2009*; *Eldar et al., 2003*).

The surprising dispensability of positive feedback for gradient formation parallels our findings on the role of negative feedback in Nodal patterning (*Rogers et al., 2017*). In that work, we showed that Lefty-mediated feedback—despite its extensive conservation across animals—was dispensable for normal development in zebrafish. Lefty was instead required for robustness; intact feedback loops enabled the embryo to correct exogenous perturbations to signaling. This raises the intriguing possibility that Nodal positive feedback serves a similar purpose. Though dispensable for gradient

formation per se, positive feedback may help to ensure that a gradient of the appropriate shape and intensity forms even in the face of mutations, environmental insults or signaling noise.

# Materials and methods

## Key resources table

| Reagent type (species) or resource | Designation | Source or reference | Identifiers | Additional information |
|---|---|---|---|---|
| Gene (*Danio rerio*) | *oep* (*tdgf1*) | ZFIN | ZDB-GENE-990415–198 | |
| Gene (*Danio rerio*) | *ndr1* (*sqt*) | ZFIN | ZDB-GENE-990415–256 | |
| Gene (*Danio rerio*) | *ndr2* (*cyc*) | ZFIN | ZDB-GENE-990415–181 | |
| Gene (*Danio rerio*) | *smad2* | ZFIN | ZDB-GENE-990603–7 | |
| Gene (*Danio rerio*) | *vg1* (*gdf3*) | ZFIN | ZDB-GENE-980526–389 | |
| Gene (*Danio rerio*) | *lft1* | ZFIN | ZDB-GENE-990630–10 | |
| Gene (*Danio rerio*) | *lft2* | ZFIN | ZDB-GENE-990630–11 | |
| Strain, strain background (*Danio rerio*) | AB | ZIRC | ZDB-GENO-960809–7 | |
| Strain, strain background (*Danio rerio*) | TL | ZIRC | ZDB-GENO-990623–2 | |
| Genetic reagent (*Danio rerio*) | *oep*$^{tz57}$ | *Hammerschmidt et al., 1996* | RRID:ZDB-ALT-980203-1256 | |
| Genetic reagent (*Danio rerio*) | *sqtt*$^{cz35}$ | *Feldman et al., 1998* | RRID:ZDB-ALT-000913-2 | |
| Genetic reagent (*Danio rerio*) | *cyc*$^{m294}$ | *Sampath et al., 1998* | RRID:ZDB-ALT-980203-609 | |
| Genetic reagent (*Danio rerio*) | *smad2*$^{vu99}$ | *Dubrulle et al., 2015* | RRID:ZDB-ALT-150807-1 | |
| Genetic reagent (*Danio rerio*) | *vg1*$^{a165}$ | *Montague and Schier, 2017* | RRID:ZDB-ALT-180515-7 | |
| Genetic reagent (*Danio rerio*) | *lft1*$^{a145}$ | *Rogers et al., 2017* | RRID:ZDB-ALT-180417-4 | |
| Genetic reagent (*Danio rerio*) | *lft2*$^{a146}$ | *Rogers et al., 2017* | RRID:ZDB-ALT-180417-5 | |
| Recombinant DNA reagent | *pJZoep Flag1-2* | *Zhang et al., 1998* | | Template for in vitro transcription |
| Recombinant DNA reagent | *SV40NLS-sfgfp* in pCS2+ | Gift from Dr. Jeffrey Farrell | | Template for in vitro transcription |
| Recombinant DNA reagent | *sqt-sfGFP* in pCS2+ | *Montague and Schier, 2017* | | Template for in vitro transcription |

*Continued on next page*

*Continued*

| Reagent type (species) or resource | Designation | Source or reference | Identifiers | Additional information |
|---|---|---|---|---|
| Recombinant DNA reagent | *vg1-halotag* in pCS2+ | This study | Plasmid can be obtained by reaching out to N.L. | Template for in vitro transcription. |
| Recombinant DNA reagent | *sqt* in pCS2+ | *Müller et al., 2012* | | Template for in vitro transcription |
| Antibody | Anti-phospho-smad2/3 (rabbit monoclonal) | Cell Signaling Technology | #18338 | 1:1000 dilution |
| Antibody | Anti-GFP (chicken monoclonal) | Aves Labs | (RRID:AB_2307313) | 1:1000 dilution |
| Antibody | Anti-eCdh1 (mouse monoclonal) | BD Biosciences | #610181 (RRID:AB_397580) | 1:100 dilution |
| Antibody | Goat α-rabbit Alexa 647 conjugate (goat monoclonal) | Thermo-Fisher Scientific | A-21245 (RRID:AB_2535813) | 1:2000 dilution |
| Antibody | Goat α-chicken Alexa 488 (goat monoclonal) | Thermo-Fisher Scientific | A-11039 (RRID:AB_142924) | 1:2000 dilution |
| Antibody | Goat α-mouse IgG (H + L)-Alexa 488 (goat monoclonal) | Thermo-Fisher | A-32723 (RRID:AB_2633275) | 1:750 dilution |
| Peptide, recombinant protein | Pronase | Millipore Sigma | 53702 | |
| Commercial assay or kit | mMessage mMachine Sp6 kit | Thermo-Fisher | AM1340 | |
| Commercial assay or kit | E.Z.N.A. Cycle Pure | Omega Bio-Tek | D6492-01 | |
| Commercial assay or kit | E.Z.N.A. Total RNA Kit I | Omega Bio-Tek | R6834-01 | |
| Chemical compound, drug | kDa Alexa488-dextran conjugate | Thermo-Fisher | D34682 | |
| Chemical compound, drug | Janelia Fluor HaloTag Ligand 646 | Promega | GA1120 | |
| Software, algorithm | ImageJ/FIJI | ImageJ/FIJI | RRID:SCR_002285 | Image Analysis |
| Software, algorithm | MATLAB | Mathworks | RRID:SCR_001622 | Image Analysis, Simulations |

## Genotyping

Genomic DNA was isolated via the HOTSHOT method from either excised adult caudal fin tissue or individual fixed embryos (*Meeker et al., 2007*). Genotyping was carried out via PCR under standard conditions followed by restriction enzyme digest when appropriate. For brevity, allele designations were omitted in the rest of the text. *lefty1*[a145]: The *lefty1*[a145] allele contains a 13-base-pair deletion that destroys a PshAI restriction site and was detected as in *Rogers et al., 2017*. *lefty2*[a146]: The *lefty2*[a146] allele contains an 11-base-pair deletion and was detected as described (*Rogers et al., 2017*). squint[cz35] : The *squint*[cz35] allele contains a ~ 1.9 kb insertion and was detected as in *Feldman et al., 1998*. cyclops[m294] : The *cyclops*[m294] allele contains a single nucleotide polymorphism (SNP) that destroys an AgeI restriction site and was detected as described (*Sampath et al., 1998*). oep[tz57] : The *oep*[tz57] allele contains a SNP that introduces a Tsp45I restriction site

(*Zhang et al., 1998*; *Hammerschmidt et al., 1996*). The allele was detected via PCR amplification with primers *AC102* and *AC103* flanking the SNP followed by Tsp45I digestion overnight. A wild-type allele yields an undigested band of 285 bp, while a mutant allele yields bands of 140 bp and 145 bp. $vg1^{a165}$: The $vg1^{a165}$ allele contains a 29 bp deletion and was detected as described (*Montague and Schier, 2017*). $smad2^{vu99}$: The $smad2^{vu99}$ allele contains a SNP that introduces a BtsCI restriction site (*Dubrulle et al., 2015*). The allele was detected via PCR amplification with primers *NL-89* and *NL-91* flanking the SNP followed by BtsCI digestion overnight. A wild-type allele yields an undigested band of 298 bp, while a mutant allele yields bands of 221 bp and 77 bp.

## Zebrafish husbandry and fish lines

Fish were maintained per standard laboratory procedures (*Westerfield, 1993*). Embryos were raised at 28.5°C in embryo medium (250 mg/L Instant Ocean salt, 1 mg/L methylene blue in reverse osmosis water adjusted to pH seven with NaHCO₃) and staged according to a standard staging series (*Kimmel et al., 1995*). Wild-type fish and embryos represent the TLAB strain. *Lefty1, lefty2, squint, cyclops, oep,* and *vg1* mutant fish were maintained as previously described (*Montague and Schier, 2017*; *Rogers et al., 2017*; *Zhang et al., 1998*; *Feldman et al., 1998*; *Sampath et al., 1998*). $Cyc^{+/-};oep^{-/-}$, and $sqt^{+/-};oep^{-/-}$ double mutants were generated by incrossing $cyc^{+/-};oep^{+/-}$ or $sqt^{+/-};oep^{+/-}$ respectively and rescuing them with an injection of 55 pg *oep* mRNA at the one-cell stage. $Smad2^{-/-}$ germline carrier fish were obtained by germline transplantation, using $Smad2^{+/-}$ incross progeny as germ cell donors (*Ciruna et al., 2002*). Germline carrier embryos were obtained by either incrossing EK fish or crossing $dmrt1^{E3ins-/-}$ female fish to $dmrt1^{E3ins-/+}$ male fish. The $dmrt1^{E3ins-/-}$ and $dmrt1^{E3ins-/+}$ fish were gifts from Kaitlyn A. Webster/Kellee R. Siegfried and were used with the intent of biasing germline carriers to female adult fates (*Webster et al., 2017*).

For experiments shown in the text, mutant embryos were derived as follows: *MZoep* embryos were obtained by crossing $oep^{-/-}$ adults; *Zoep* embryos were obtained by crossing $oep^{\pm}$ females with $oep^{-/-}$ males (see genotyping below); *MZsmad2* embryos were obtained by crossing $smad2^{-/-}$ germline carrier adults; *Mvg1* embryos were obtained by crossing $vg1^{-/-}$ females with TLAB males; $lft1^{-/-};lft2^{-/-}$ embryos were obtained by crossing $lft1^{-/-};lft2^{-/-}$ adults; $sqt^{+/+};MZoep, sqt^{+/-};MZoep,$ and $sqt^{-/-};MZoep$ embryos were obtained by crossing $sqt^{+/-};oep^{-/-}$ adults; $cyc^{+/+};MZoep, cyc^{+/-};MZoep,$ and $cyc^{-/-};MZoep$ embryos were obtained by crossing $cyc^{+/-};oep^{-/-}$ adults.

## mRNA synthesis and microinjection

pCS2 +vectors containing the CDS of either *SV40NLS-sfgfp* or *oep* were linearized with NotI and subsequently purified with the E.Z.N.A. Cycle Pure (Omega) kit. Purified templates were transcribed using the mMESSAGE mMACHINE SP6 (Invitrogen/Thermo Fisher Scientific) kit, and the resulting *gfp* and *oep* capped mRNAs were purified with the E.Z.N.A. Total RNA Kit I (Omega). Capped mRNA concentrations were evaluated via NanoDrop (Thermo Fisher Scientific) spectrophotometry. Kits were used per manufacturer's respective protocols.

## Sensor cell transplant experiments

M*vg1* sensor donors were injected with either 1 nl of 55 pg/nl *gfp* mRNA or 1 nl of 55 pg/nl *gfp* mRNA +110 pg/nl *oep* mRNA (**Figure 3B**) at the one-cell stage. *MZsmad2* +*oep* hosts (**Figure 3D**) were injected with 1 nl of 110 pg/nl *oep* mRNA at the one-cell stage. Prior to injection, both donor and host embryos were enzymatically dechorionated using 1 mg/ml Pronase (Millipore Sigma). After injection, embryos were raised at 28.5°C in 1% agarose-coated plastic dishes in embryo medium. At high stage, donor and host embryos were placed in 1X Danieau's buffer, and ~5–10 blastomeres were transplanted from the animal pole of donor embryos to the margin of host embryos, unless specified otherwise. After transplantation, host embryos were returned to embryo medium and raised to 50% epiboly at 28.5°C before fixation.

## α-pSmad2 immunostaining

The protocol was modified from *Rogers et al., 2017*. Briefly, embryos were fixed in 4% formaldehyde overnight at 4°C in 1x PBSTw (1x PBS + 0.1% (v/v) Tween 20), washed in 1x PBSTw, dehydrated in a MeOH/PBST series (25%, 50%, 75%, and 100% MeOH), and stored at −20°C until staining. Embryos were rehydrated in a MeOH/PBSTr (1x PBS + 1% (v/v) Triton X-100) series (75%, 50%, and

25% MeOH), washed 3x in PBSTr, and manually de-yolked. Embryos were then incubated for 2 hr at room temperature (RT) in antibody binding buffer (PBSTr +1% (v/v) DMSO) before overnight incubation with 1:1000 α-pSmad2 antibody (Cell Signaling Technology #18338) and, when required, 1:1000 α-GFP antibody (Aves Labs AB_2307313) in antibody binding buffer at 4°C. After 1° antibody incubation, embryos were washed 6X with PBSTr before a 30 min RT incubation in antibody binding buffer. Embryos were then incubated in 1:2000 goat α-rabbit Alexa 647 conjugate (ThermoFisher A-21245) and, when required, 1:2000 goat α-chicken Alexa 488 conjugate (ThermoFisher A-11039) in antibody binding buffer. Embryos were then washed 6X with PBSTr and 1X PBSTw respectively before a 30 min RT incubation with DAPI. Embryos were washed 3X in PBSTr before dehydration in a MeOH/PBSTw series (50% and 100% MeOH). Embryos were stored at −20°C in MeOH until imaging.

## Embryo clearing and imaging

Embryos were first cleared in 2:1 benzyl benzoate:benzyl alcohol (BBBA) (*Yokomizo et al., 2012*). After clearing, embryos were mounted in BBBA in individual wells of a 15-well multitest slide (MP Biomedicals). Mounting was performed under a Zeiss Stemi 2000 stereoscope fitted with a Nightsea adaptor system with UV filters and light head to enable embryo visualization. Embryos were then cracked with forceps before placement of a #1.5 coverslip, approximately flattening the embryos. The coverslip was secured with adhesive tape before imaging on a Zeiss LSM-700 inverted confocal microscope.

## smFISH probe synthesis

Single-molecule fluorescent in situ hybridization (smFISH) probes against the coding sequences of *cyclops* and *squint* were designed using the Stellaris Probe Designer, with oligo length 18–22 bp and minimum spacing length two nucleotides. Probes were then checked for cross-reactivity between orthologs (probes with <4 mismatches were discarded) and ordered with 3′ C7 amino group modifications (IDT). Thirty-nine probes against *cyclops* and 44 against *squint* were purchased. Probe libraries for each gene were pooled, dehydrated in a Speedvac, and resuspended in water at a concentration of 1 mM. Probes were then coupled to Atto-647N NHS ester (Millipore Sigma #18373) per supplier protocol and purified with the Zymo Oligo Clean and Concentrator kit. Probe concentration was then determined using NanoDrop (Thermo Fisher Scientific) spectrophotometry.

## smFISH staining and imaging

The smFISH staining protocol is modified from previous reports (*Oka and Sato, 2015*; *Stapel et al., 2016*). Briefly, embryos were fixed in 4% formaldehyde overnight at 4°C in 1x PBSTw (1x PBS + 0.1% (v/v) Tween 20), washed in 1x PBSTw, dehydrated in a MeOH/PBST series (50% and 100% MeOH), and stored at −20°C until staining. Embryos were rehydrated in a MeOH/PBSTw (50% and 100% PBSTw) series before manual deyolking. Embryos were then incubated in pre-hybridization buffer (preHB) (10% formamide, 2x SSC, 0.1% (v/v) TritonX-100), 0.02% (w/v) BSA, and 2 mM ribonucleoside-vanadyl complex (NEB) for 30 min at 30°C before overnight incubation with 10 nM probes in hybridization buffer (10% (w/v) dextran sulfate (MW 500,000) in preHB) at 30°C in the dark. After staining, embryos were washed 2 × 30 min at 30°C in hybridization wash solution (10% (v/v) formamide, 2x SSC, 0.1% (v/v) Triton X-100) before a brief wash in 2x SSC +0.1% (v/v) Tween-20. Finally, embryos were incubated for 20 min at 30°C in 0.2X SSC before a 15-min incubation in DAPI and 2 × 2 x SSC +0.01% Tween washes.

For membrane staining, 1:100 α-eCdh1 antibody (BD Biosciences #610181) was added overnight with the probes in hybridization buffer. After the 20 min 0.2X SSC wash, 1:750 Goat α -mouse IgG (H + L)-Alexa 488 (ThermoFisher A32723) in PBSTw was added, and embryos were incubated for 2 hr at RT in the dark. Embryos were washed 6X with PBSTw before a 15 min DAPI incubation and 2 × 2 x SSC +0.01% Tween washes.

For mounting, embryos were kept in 2X SSC, cut from the margin to the animal pole with a scalpel, and mounted in 2X SSC on a standard glass slide between two double-sided adhesive tape bridges (3M Scotch). A #1.5 coverslip then approximately flattens the embryo and is secured in place by the adhesive tape. Mounted embryos were then imaged on a Zeiss LSM-880 inverted confocal using the Airyscan detector.

## Image segmentation

Staining intensities for individual nuclei were compiled for *Figures 1–3*. Nuclei were segmented from DAPI channel images using a custom pipeline implemented in MATLAB as described previously (*Rogers et al., 2017*). Before segmentation, each image stack was manually inspected to identify acceptable z-bounds. Lower bounds were chosen to exclude internal YSL nuclei from the segmentation. Briefly, for each slice, out-of-plane background signal was approximated by blurring adjacent Z-slices with a Gaussian smoothing kernel and subtracted. Nuclei boundaries were identified using an adaptive thresholding routine (http://homepages.inf.ed.ac.uk/rbf/HIPR2/adpthrsh.htm). Spurious objects were discarded by morphological filtering (area threshold followed by image opening with a disc-shaped structuring element).

Three-dimensional objects were compiled from the two-dimensional segmentation results with a simple centroid-matching scheme. A disc of diameter five pixels was defined centered at the centroid of each two-dimensional object, and three-dimensional objects were identified by object labeling with a 6-connected neighborhood. Intuitively, this procedure matches objects whose centroids are separated by <10 pixels (i.e. twice the disc diameter used prior to object matching). Objects that fail to span at least 2 Z-slices were discarded. Fluorescence intensities in the DAPI, GFP and pSmad2 channels were compiled as average pixel intensities within the three-dimensional segmentation boundaries.

## Genotyping of Z*oep, cyc;oep,* and *sqt;oep* mutant embryos

Crosses leading to homozygous Z*oep, cyc;oep,* and *sqt;oep* mutant embryos were generated from non-homozygous parents. Specifically, Z*oep* embryos were generated by crossing an $oep^{-/-}$ male against a $oep^{+/-}$ female; *cyc;oep* embryos were generated from a cross between $cyc^{+/-};oep^{-/-}$ parents; *sqt;oep* embryos were generated from a cross between $sqt^{+/-};oep^{-/-}$ parents. To identify the genotype of embryos used for imaging, each embryo was manually cut into halves (i.e. through the animal pole) with a clean scalpel after pSmad2 immunostaining. One half of the embryo was dehydrated for clearing and imaging (as described in the α-pSmad2 immunostaining methods section), and the other was used for genomic DNA preparation and genotyping. Genotyping was carried out for each mutation as summarized above. For Z*oep* staining, genotyping was carried out as described for 30% epiboly and 50% epiboly stages; this revealed that Z*oep* embryos could be clearly identified by average staining intensity. Shield-stage Z*oep* embryos were identified by staining intensity.

## Sensor cell identification and gradient quantification

All gradient quantifications in *Figures 1–3* plot nuclear staining intensity as a function of distance from the embryonic margin. Because the margin boundary is curved in our flat mounts, these distances are not a simple function of position within the image. A semi-automated routine was therefore implemented in MATLAB to calculate the distance from the margin for each nucleus. In brief, the YSL-embryo boundary was manually identified and drawn using maximum intensity projections of the DAPI channel. This boundary was then converted into a binary mask and a distance transform was applied. After the distance transform, every pixel in the image adopts a value equal to its distance to the closest non-zero pixel (i.e. the margin contour); the distance from the margin for each nucleus was defined as the pixel intensity of the distance transform image at the corresponding centroid position.

In order to quantify the gradients in Nodal-insensitive host embryos, sensor cells had to be specifically identified. A classification scheme based on nuclear GFP intensity was therefore devised. Because there was some background α-GFP staining, even in cells that did not receive *gfp* mRNA, the approximate baseline GFP intensity was identified by taking a sliding window median of GFP staining intensity as a function of nuclear distance from the margin. GFP$^{+}$ cells were identified as having nuclei brighter than 3-fold above the local baseline, and GFP$^{-}$ cells were identified as having staining intensity at or below the local baseline. These thresholds are stringent and resulted in some false-negative nuclear classifications (e.g. likely GFP$^{+}$ nuclei that failed to be classified as such). However, they do ensure that the nuclei plotted in the main text represent *only* clear GFP$^{+}$ or GFP$^{-}$ populations. This analysis was also performed using less stringent thresholds and manual correction of results, which generated comparable conclusions to the results presented in the paper.

After calculation of GFP staining status and distance from the margin for each nucleus, average gradients were compiled. To facilitate comparison between replicate embryos, the pSmad2 staining intensities were normalized to the baseline intensity (i.e. average nuclear intensity of all nuclei falling between 150 and 200 µm) from the margin. After this normalization, data from each embryo were pooled, and average gradients were compiled with a sliding window average (solid curves in quantified gradients in *Figures 1–3*) with a window size of 20 µm. Due to sparse sampling of the gradients by sensor cells, some statistical fluctuations in average gradient shape are evident (e.g. the 'hump' in *Figure 2C*).

## YSL-specific expression and visualization of Halo-tagged Vg1-Squint heterodimers

In *Figure 3—figure supplement 1*, visualization of Halo-tagged Vg1-Squint heterodimers was achieved using a modified sensor cell assay. Donor and host embryos were collected in 1X E3 medium at the one-cell stage and immediately dechorionated with 1 mg/ml Pronase (Protease type XIV from *Streptomyces griseus*, Millipore Sigma). Host embryos for the MZ*oep +oep* mRNA condition were injected with 110 pg *oep* mRNA at the one-cell stage. Wild-type donor embryos were injected with 110 pg oep mRNA and 0.6 ng of 3 kDa Alexa 488-dextran conjugate (Invitrogen) at the one-cell stage. Sensor cells derived from *oep*-injected donors were used to enhance the sensitivity of the assay. As increased *oep* expression improves ligand capture (*Figure 2B*), fluorescent ligand accumulation was easier to observe using these sensors.

Between the 512-cell stage and the 1 k-cell stage, when the YSL has become an obvious structure, the YSL of host embryos was injected at four adjacent points with 0.065 nl of 255 ng/µl *vg1-HaloTag* mRNA, 167 ng/µl *squint* mRNA, and 90 ng/µl *gfp* mRNA. Injected host embryos were left to recover in 1X E3 medium at 28.5°C for 20 min before being transferred to 1.5 mL non-stick surface micro-centrifuge tubes (VWR) filled with 250 µl of 200 nM Janelia Fluor HaloTag Ligand (Promega) that had been diluted in 1X E3 medium. Tubes were then placed at a shallow ~10° angle and left in the dark at 28.5°C for 1 hr.

After staining, embryos were transferred into 1X Ringer's solution (116 mM NaCl, 2.8 mM KCl, 1 mM CaCl2, 5 mM HEPES) for transplantation. Cells were transplanted from the animal pole of *oep*-injected wild-type donor embryos to the animal pole of host embryos. After transplantation, host embryos were left to recover in 1X Ringer's solution for 10 min before being transferred back to 1.5 mL non-stick surface micro-centrifuge tubes (VWR) filled with 250 µl of 200 nM Janelia Fluor HaloTag Ligand (Promega) that had been diluted in 1X E3 medium. Tubes were again placed at a shallow ~10° angle and left in the dark at 28.5°C.

Thirty min before imaging, embryos were removed from HaloTag ligand solution and washed 2 × 7 min in 1X E3 medium in separate wells of a 1% agarose-coated plastic 6-well plate. After washing, embryos were mounted in 1% low melting point agarose on glass-bottom Petri dishes (VWR) with animal poles facing the coverslip. After initial agarose droplets containing embryos had set, the entire coverslip was covered in 1% low melting point agarose and subsequently covered in 1X E3 medium after setting.

Embryos were imaged on a confocal microscope with Olympus IX83 stand, UPL S APO 30x silicon oil objective, Yokogawa CSU-W1 confocal scanner unit, and Hamamatsu ORCA-Fusion camera. Adjacent Z slices were taken with 3 µm spacing.

## Visualization of Squint-sfGFP gradients with transplanted source cells

In *Figure 2—figure supplement 3*, ectopic Squint-sfGFP gradients were generated in wild-type, MZ*oep* and *oep*-overexpressing MZ*oep* hosts using transplanted source cells. This was carried out as described previously (*Müller et al., 2012*). Briefly, wild-type donors were injected with 250 pg *sqt-sfgfp* mRNA and 0.2 ng of 10 kDa Alexa647-dextran conjugate (ThermoFisher) at the one-cell stage. MZ*oep +oep* hosts were injected with 110 pg *oep* mRNA at the one-cell stage. All embryos were stored in 1% agarose-coated plastic plates in 1X E3 medium at 28.5°C after injection.

At sphere stage, embryos were transferred to 1X Ringer's solution for transplantation. Approximately 50 source cells were removed from donor embryos and left briefly in 1X Ringer's solution to allow for the dissipation of any cellular debris and residual secreted ligand before approximately transplantation to the animal pole of host embryos. Embryos were left in 1X Ringer's solution for 10

min to recover before being transferred to 1X E3 medium in 1% agarose coated plates at 28.5°C in the dark. After 1 hr and 40 min from the time of the last transplantation, embryos were mounted in 1% low-melting point agarose on glass-bottom Petri dishes and imaged by confocal microscopy as described in the previous section.

Gradients were quantified as previously described (*Müller et al., 2012*). Briefly, maximum intensity projections comprising 15 consecutive confocal slices were prepared for each embryo. A region of interest adjacent to the transplanted clone (approximately 150 µm long by 40 µm wide) was extracted from each projection image. Average background intensities from ungrafted control embryos were subtracted, fluorescence intensities were averaged within 1 µm bins, and curves were smoothed by sliding window averaging (window size of 5 µm). The resulting curves were sampled every 5 µm. Each curve was normalized to the fluorescence intensity immediately adjacent to the source.

### Kymograph preparation in *Figure 5D and E*

In the experimental section of *Figure 5*, kymographs were presented that average the behavior of replicate embryos (bars to the right of representative images in *Figure 5D and E*). To prepare these kymographs, the distance from the margin for each pixel in the maximum intensity projection α-pSmad2 image was calculated as described in the above section. Pixels were then binned by distance from the margin and averaged across embryos to generate the plots in *Figure 5*. Each vertical bar in the plot was drawn for all of the data from a given stage (from left to right: dome, 50% epiboly and shield). Color scalings were selected for visibility and are not equivalent between the wild-type and Z*oep* datasets.

### Gradient simulations

Sensor cell assay simulations were implemented in MATLAB using the PDE toolbox. Simulations were carried out on a two-dimensional rectangular slab (100 × 300 µm) with no-flux boundary conditions. The Nodal source was simulated as a thin strip of tissue (the first 5 µm) that produced Nodal at a constant rate. Sensor cells were simulated as small circular domains with permeable boundaries (6 µm diameter) in which parameters (e.g. presence or absence of free receptors) could be set independently of the rest of the tissue. Simulations were run ~2.5 hr of simulation time in an effort to mimic the normal duration of Nodal gradient spread in zebrafish embryos. Simulations are described in detail in the SI (*Reproduction of sensor cell assay with gradient simulations*). Plots in *Figure 4* depict the entire tissue domain at the conclusion of the simulations.

Simulations incorporating receptor production and replacement were implemented in MATLAB using pdepe. Simulations were carried out on a one-dimensional tissue (300 µm long) with no-flux boundary conditions. The Nodal source was simulated as a thin strip of tissue (the first 5 µm) that produced Nodal at a constant rate. Simulations were run for ~2.5 hr of simulation time in an effort to mimic the normal duration of Nodal gradient spread in zebrafish embryos. Simulations are described in detail in the SI (*Gradient simulations accounting for receptor production and consumption*). Plots in *Figure 5B and C* are kymographs summarizing the state of the system at regularly sampled times. Each column of kymograph shows the concentration of a given component at each position in the system ('YSL' at the bottom), and adjacent columns are separated by 20 s of simulation time. Kymographs begin plotting data at t = 0 to capture the transients associated with gradient formation. Pixel scalings (i.e concentration scales) are not identical between *Figure 5b and c*; scalings were chosen to maximize data visibility. Due to the absence of receptor replacement, concentrations of free receptor and receptor-ligand complexes are markedly lower in *Figure 5c* (in accordance with experimental data in Z*oep* mutants, see *Figure 5F*).

## Acknowledgements

This research was supported by the National Institutes of Health (R37GM056211 to AFS, K99-HD097297-01 to NDL, T32GM080177 training grant supported ANC), the Arnold and Mabel Beckman Foundation (postdoctoral fellowship to NDL), the NSF (GRFP DGE1745303 to ANC), a Simmons Family Imaging Award (to ANC), and the Damon Runyon Cancer Research Foundation (postdoctoral fellowship to PBA). NDL, ANC and AFS conceived the project and designed experiments; NDL and ANC performed the experimental work and analysis. NDL, ANC and AFS wrote the paper. PBA

identified the pSmad2 antibody used for immunostaining. We thank Kaitlyn Webster and Kellee Siegfried for the generous gift of *dmrt1* mutant zebrafish. We thank Doug Richardson, the Harvard Center for Biological Imaging, and the Biozentrum Imaging Core Facility for microscopy infrastructure and support. We thank Jeffrey Farrell, Katherine Rogers, Harold McNamara, and PC Dave P Dingal for helpful comments on the manuscript.

## Additional information

### Funding

| Funder | Grant reference number | Author |
| --- | --- | --- |
| National Institutes of Health | K99-HD097297-01 | Nathan D Lord |
| National Institutes of Health | R37GM056211 | Alexander F Schier |
| National Institutes of Health | T32GM080177 | Adam N Carte |
| National Science Foundation | DGE1745303 | Adam N Carte |
| Arnold and Mabel Beckman Foundation | | Nathan D Lord |
| Damon Runyon Cancer Research Foundation | | Philip B Abitua |

The funders had no role in study design, data collection and interpretation, or the decision to submit the work for publication.

### Author contributions

Nathan D Lord, Conceptualization, Resources, Data curation, Software, Formal analysis, Funding acquisition, Investigation, Methodology, Writing - original draft, Writing - review and editing, Conceived the project, Designed experiments, Performed experimental work and analysis, Wrote the paper; Adam N Carte, Conceptualization, Data curation, Software, Formal analysis, Investigation, Methodology, Writing - original draft, Writing - review and editing, Conceived the project, Designed experiments, Performed experimental work and analysis, Wrote the paper; Philip B Abitua, Methodology, Identified the pSmad2 antibody used for immunostaining; Alexander F Schier, Conceptualization, Supervision, Funding acquisition, Writing - review and editing, Conceived the project and designed experiments, Wrote manuscript

### Author ORCIDs

Nathan D Lord (iD) https://orcid.org/0000-0001-9553-2779
Adam N Carte (iD) https://orcid.org/0000-0002-3791-4872
Philip B Abitua (iD) https://orcid.org/0000-0002-8264-0807
Alexander F Schier (iD) https://orcid.org/0000-0001-7645-5325

### Ethics

Animal experimentation: All vertebrate animal work was performed at the facilities of Harvard University, Faculty of Arts & Sciences (HU/FAS). The HU/FAS animal care and use program maintains full AAALAC accreditation, is assured with OLAW (A3593-01), and is currently registered with the USDA. This study was approved by the Harvard University/Faculty of Arts & Sciences Standing Committee on the Use of Animals in Research & Teaching under Protocol No. 25-08.

### Decision letter and Author response

Decision letter https://doi.org/10.7554/eLife.54894.sa1
Author response https://doi.org/10.7554/eLife.54894.sa2

## Additional files

### Supplementary files

• Supplementary file 1. Single-molecule FISH probe set sequences. Measurement of expression of *cyclops* and *squint* mRNA by smFISH in wild type, MZ*oep* and MZ*smad2* backgrounds is presented in *Figure 1—figure supplement 2*. Sequences for the requisite smFISH probe sets are organized in this table. Details of probe-dye coupling and embryo staining are contained in the Materials and Methods section of the manuscript.

• Supplementary file 2. PCR primer sequences. This table organizes the PCR primers used for mutant genotyping. Genotyping protocols are described in detail in the Materials and methods section.

• Transparent reporting form

### Data availability

Source data files have been provided for all quantified immunofluorescence datasets.

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

# Appendix 1

## Computational Models of Nodal Gradient Formation

### Reproduction of sensor cell assay with gradient simulations

Simulations presented in *Figure 4* were implemented using the MATLAB PDE toolbox. The host embryo was represented as a two-dimensional rectangular slab (100 × 300 μm). 'Sensor cells' were simulated as circular domains of 6 μm diameter—with independently set simulation parameters—scattered throughout the rectangular domain. Equations governing the model are specified as below:

$$\frac{\partial N(x,t)}{\partial t} = D_N \nabla^2 N(x,t) - k_1 N(x,t)R + k_{-1}C(x,t)$$

$$\frac{\partial C(x,t)}{\partial t} = k_1 N(x,t)R - k_{-1}C(x,t) - k_2 C(x,t)$$

$$\frac{\partial S(x,t)}{\partial t} = k_s C(x,t) - k_{-s}S(x,t)$$

Where $N(x,t)$, $C(x,t)$, and $S(x,t)$ refer to the concentration of free Nodal, Nodal-Receptor complex, and pSmad2 at position $x$ at time $t$, respectively. Parameter values are summarized in the table below. All boundaries are specified as no-flux, and $N(x,t=0)=0$ and $S(x,t=0)=0$ were assumed for initial conditions. The Nodal source was simulated by specifying a constant Nodal production rate ($\lambda_N$) for points lying in the region $0 \leq x \leq 5$. For simplicity, we assume the receptor concentration, $R$, to be constant at each position throughout the simulation. Simulations were run for ~2.5 hours of simulation time to mimic the normal duration of Nodal spread in zebrafish embryos.

Remarks:

1. These simulations instantiate a simple model of morphogen gradient formation—constant synthesis at a localized source coupled with linear degradation—that has been discussed at length elsewhere (*Rogers and Schier, 2011*; *Wartlick et al., 2009*; *Zhou et al., 2012*; *Lander et al., 2002*). The steady state gradient for this is a single exponential with length scale set by the diffusion constant and effective Nodal degradation rate (i.e. through complexing with the receptor). Small deviations from this expectation arise due to the finite size of the Nodal source domain.

2. For the plots in the main text, we track signaling through $S$, the concentration of phosphorylated Smad2. We assume the phosphorylation rate to be first-order with respect to the ligand-receptor complex—effectively that unphosphorylated Smad2 is neither depletable nor present in high enough concentrations to reveal saturation of receptor complex kinase activity—and the dephosphorylation rate to be first order with respect to $S$. Intuitively, this means that $S$ reflects receptor occupancy over a time window of $\sim 1/k_{-s}$. Signaling could also be tracked as the concentration of occupied receptor and yields qualitatively similar results.

3. For simulations of sensor cells transplanted into MZ*oep* hosts (*Figures 2B*, *4C*), the receptor concentration ($R$) was set to zero outside of the sensor cell boundaries. Within these boundaries, $R$ was kept at the 'wild-type' level (i.e. at the same levels as in the simulations from *Figure 4b*). We note that the increased signaling intensity in this background reflects the fact that binding with receptor is the only available degradation pathway for the ligand; when $R=0$, available ligand concentrations are substantially higher throughout the "embryo".

4. For simulations of sensor cells transplanted into *oep*-overexpressing hosts (i.e. MZ*smad2* + *oep* mRNA, *Figures 3d*, *4d*), $R$ was increased by a factor of 2 throughout the host regions, and $k_s$ was set to zero to mimic the absence of Smad2. Simulation parameters within the sensor cell regions retained their 'wild-type' values.

## Gradient simulations accounting for receptor production and consumption

The model presented in *Figure 5* explicitly accounts for production and consumption of receptor components. For the figure panels, the model was simulated on a 1-dimensional domain of length 300 μm. Model equations were specified as follows:

$$\frac{\partial N(x,t)}{\partial t} = D_N \nabla^2 N(x,t) - k_1 N(x,t) R(x,t) + k_{-1} C(x,t)$$

$$\frac{\partial R(x,t)}{\partial t} = k_3 - k_1 N(x,t) R(x,t) + k_{-1} C(x,t) - k_{-R} R(x,t)$$

$$\frac{\partial C(x,t)}{\partial t} = k_1 N(x,t) R - k_{-1} C(x,t) - k_2 C(x,t)$$

Here, $N(x,t)$, $R(x,t)$, and $C(x,t)$ refer to the concentration of free Nodal, free receptor and receptor-Nodal complexes, respectively, at position $x$ and time $t$. No-flux boundary conditions were assumed at both ends of the domain, and Nodal was produced at a constant rate $\lambda_N$ within a source domain covering $0 \leq x \leq 5$. $N(x, t = 0)$ and $C(x, t = 0)$ were initiated at zero throughout the field, and $R(x, t = 0)$ was set to $\lambda_R / k_{-R}$ for all positions. Simulations were implemented in MATLAB using pdepe.

1. In this model, receptor synthesis is treated as constitutive throughout the field, and degradation occurs through ligand-dependent and ligand-independent mechanisms. The ligand-dependent pathway occurs through degradation of receptor-ligand complexes (schematically represented as endocytosis in *Figures 4* and *5*, rate $k_2 C(x,t)$). The ligand-independent pathway is assumed to be first-order with respect to free receptor (rate $k_{-R} R(x,t)$).

2. Direct degradation of the receptor—once we have made the step of assuming constitutive production– is required to achieve steady state concentrations of free receptor outside the domain of ligand diffusion. In the absence of this degradation route, receptor levels increase without bound far from the source as the synthesis term is not coupled to receptor levels. We note that this constitutive degradation explains the gradual decrease of free receptor levels far from the source in the *Figure 4C*; without replacement, ligand-independent degradation would eventually remove all of the receptor in the system. We include this degradation mechanism in the 'Z*oep*' simulations for parity, however the signaling wave occurs even with $k_{-R}$ set to 0.

3. Ligand-dependent removal of receptor is the key requirement for appearance of the signaling wave in Z*oep* mutants. While we regard ligand-dependent endocytosis of receptor complexes as biologically plausible, our model is agnostic to the actual mechanism. Other mechanisms can be imagined; for example, irreversible inactivation of receptor-ligand complexes that remain on the cell surface could also support formation of a wave provided that the ligand does not dissociate. Indeed, we suspect that any mechanism through which ligand binding renders receptors incapable further ligand capture or signaling would support wave formation.

4. For *Figure 5B and C*, we summarize the simulation results for each component with kymographs. Each column of these images shows the state of the 1-dimensional system—with source at the bottom and 'animal pole' at the top—at each point. Simulation time proceeds from left to right, and each plot represents two hours of simulation time. We note that, with biologically reasonable parameters, we observe formation of a stable gradient with receptor replacement and clear propagation of the wave without receptor replacement.

5. We chose to use a diffusion rate of 30 μm²/sec for the Nodal ligand. We note that this rate is substantially faster than the effective diffusion rate observed by Muller et al. We chose this value to illustrate that—even for a highly diffusive ligand— short-range gradients can be generated by efficient capture. Use of the previously measured diffusion rate (~3 μm²/sec) does not compromise formation of the wave and is therefore not critical for the conclusions of the paper. Instead, this change results in a traveling wave with a 'narrower' profile, as diffusing ligand does not travel as far into a field of free receptor before capture.

## Model Parameters

| Parameter | Description | Value | Intuition for value | Reference |
|---|---|---|---|---|
| $\lambda_N$ | Nodal source production rate | 8e-6 μM/sec | ~ 3 Nodal ligand molecules/min per cell | - |
| $D_N$ | Free Nodal diffusion rate | 30 μm²/sec | MSD of ~ 650 μm after 2 hr | *Müller et al., 2012* |
| $k_1$ | Nodal-Receptor association rate | 10/μM*sec | Mean time to capture of ~ 2 s at initial Oep concentration | *De Crescenzo et al., 2003* |
| $k_{-1}$ | Nodal-Receptor dissociation rate | 6.25e-4/sec | Mean time to dissociation of 25 min | *De Crescenzo et al., 2003* |
| $k_2$ | Nodal-Receptor complex internalization rate | 1.7e-3/sec | Mean time to internalization of 10 min | *Jullien and Gurdon, 2005* |
| $k_3$ | Receptor production rate | 1.6e-5 μM/sec | Approximately 0.7 molecules/sec produced by a cell of 10 μm diameter | - |
| $k_{-R}$ | Receptor degradation rate (ligand independent) | 2.7e-4/sec | Average lifetime of 1 hr | - |
| $N(x, t = 0)$ | Free ligand initial condition | 0 μM for all x | Embryo starts out with no Nodal. | - |
| $R(x, t = 0)$ | Free receptor initial condition | 0.06 μM for all x | Approximately 2900 molecules for a cell of 10 μm diameter | *Dyson and Gurdon, 1998* |
| $C(x, t = 0)$ | Nodal-Receptor complex initial condition | 0 μM for all x | Embryo starts out with no receptor-ligand complexes. | - |

## Gradient simulations accounting for receptor trafficking

Cell culture studies have suggested that canonical TGF-β receptors are internalized at the same rate whether bound or unbound by ligand (*Anders et al., 1997*). However, the model presented in *Figure 5* assumes that the rate of receptor endocytosis increases upon ligand binding (i.e. $k_2 > k_{-R}$). As discussed in remark three in the previous section, this mechanistic assumption is not critical for wave formation. Instead, the key requirement is that ligand binding results in an increase in the effective clearance rate of receptors. For the TGF-β system, this requirement can be satisfied at the level of receptor trafficking; active receptors are 'downregulated' after endocytosis (*Mitchell et al., 2004*), while inactive receptors are recycled back to the plasma membrane. To demonstrate that this mechanism could support wave formation, we formulated a model that explicitly accounts for endosomal trafficking.

$$\frac{\partial N(x,t)}{\partial t} = D_N \nabla^2 N(x,t) - k_1 N(x,t) R_{out}(x,t) + k_{-1} C_{out}(x,t)$$

$$\frac{\partial R_{out}(x,t)}{\partial t} = k_3 - k_1 N(x,t) R_{out}(x,t) + k_{-1} C_{out}(x,t) - k_{-R} R_{out}(x,t) + k_{re} R_{in}(x,t)$$

$$\frac{\partial C_{out}(x,t)}{\partial t} = k_1 N(x,t) R_{out}(x,t) - k_{-1} C_{out}(x,t) - k_{-R} C_{out}(x,t)$$

$$\frac{\partial R_{in}(x,t)}{\partial t} = k_{-R} R_{out}(x,t) - k_{lys} R_{in}(x,t) - k_{re} R_{in}(x,t)$$

$$\frac{\partial C_{in}(x,t)}{\partial t} = k_{-R} C_{out}(x,t) - k_{lys} C_{in}(x,t) F$$

Here, $N(x,t)$, $R_{out}(x,t)$, $C_{out}(x,t)$, $R_{in}(x,t)$, and $C_{in}(x,t)$ refer to the concentration of free Nodal, free external receptor, external receptor-Nodal complexes, internalized free receptor and internalized Nodal-receptor complexes, respectively, at position $x$ and time $t$. No-flux boundary conditions were

assumed at both ends of the domain, and Nodal was produced at a constant rate $\lambda_N$ within a source domain covering $0 \leq \mathrm{x} \leq 5$. $N(x,t=0)$, $C_{out}(x,t=0)$, $R_{in}(x,t=0)$ and $C_{in}(x,t=0)$ were initiated at zero throughout the field, and $R_{out}(x,t=0)$ was set to $\lambda_R/k_{-R}$ for all positions. Simulations were implemented in MATLAB using pdepe. Example simulation results are presented in *Figure 5—figure supplement 1*.

Remarks:

1. This model assumes that the rates of internalization of free and bound receptors are identical (internalization rate constant of $k_{-R}$ in both cases). Ligand binding increases the rate of receptor clearance at the level of receptor trafficking; the rate of clearance of $C_{in}$ increases by a factor $F$ over internalized free receptor ($R_{in}$).
2. As in the model presented in the previous section, Z*oep* mutants were simulated by setting the receptor synthesis rate ($k_3$) to 0. This results in the same transformation we observed before; with receptor replacement the signaling gradient is stable, without replacement signaling propagates outward.
3. We regard the trafficking rates (k-R,kre and klys) as approximate but biologically plausible. The residence time of receptors at the plasma membrane is set by k-R (average lifetime ~1 hr), and our chosen value is consistent with measurements of TGF-β receptors in cell culture (*Mitchell et al., 2004*), as well as Activin receptors in *Xenopus* embryos (*Jullien and Gurdon, 2005*). Selections for kre and klys lead to residence times in the endosomal compartments of ~20 min for activated and inactivated receptors. The choices for these values can tune the rate of wave propagation, but are not critical for the qualitative behavior within plausible ranges of variation.

## Trafficking Model Parameters

| Parameter | Description | Value | Intuition for value | Reference |
|---|---|---|---|---|
| $\lambda_N$ | Nodal source production rate | 4e-5 μM/sec | ~15 Nodal ligand molecules/min | - |
| $D_N$ | Free Nodal diffusion rate | 30 μm²/sec | MSD of ~650 μm after 2 hr | *Müller et al., 2012* |
| $k_1$ | Nodal-Receptor association rate | 10/μM*sec | Mean time to capture of ~2 s at initial Oep concentration | *De Crescenzo et al., 2003* |
| $k_{-1}$ | Nodal-Receptor dissociation rate | 6.25e-4/sec | Mean time to dissociation of 25 min | *De Crescenzo et al., 2003* |
| $k_3$ | Receptor production rate | 1.6e-5 μM/sec | Approximately 0.7 molecules/sec produced by a cell of 10 μm diameter | - |
| $k_{-R}$ | Receptor internalization rate | 2.7e-4/sec | Average lifetime of 1 hr | - |
| $k_{re}$ | Receptor recycling rate | 8.3e-4/sec | Average residence of ~20 min in endosome before recycling to membrane | - |
| $k_{lys}$ | Lysosomal trafficking rate | 8.9e-05 | Average residence time of ~3 hr in endosome before clearance | - |
| F | Factor increase in clearance rate | 10 | Activated receptors reside in endosome for ~20 min before clearance | - |
| N(x,t = 0) | Free ligand initial condition | 0 μM for all x | Embryo starts out with no Nodal. | - |
| R(x,t = 0) | Free receptor initial condition | 0.06 μM for all x | Approximately 2900 molecules for a cell of 10 μm diameter | *Dyson and Gurdon, 1998* |
| C(x,t = 0) | Nodal-Receptor complex initial condition | 0 μM for all x | Embryo starts out with no receptor-ligand complexes. | - |

