## [Decision Letter]

**Acceptance summary:**

In developing embryos cells make fate decisions using morphogens, diffusible signaling molecules that induce concentration-dependent responses in target cells. The manuscript by Lord et al., addresses the question of how the Nodal morphogen gradient forms in developing zebrafish embryo. This work offers two main findings: first, diffusion is sufficient for the Nodal gradient formation without a relay of Nodal production; second, the co-receptor Oep shapes the Nodal gradient and restricts its range by ligand capture.

**Decision letter after peer review:**

Thank you for sending your article entitled "The pattern of Nodal morphogen signaling is shaped by co-receptor expression" for peer review at *eLife*. Your article is being evaluated by 3 peer reviewers, and the evaluation is being overseen by a Reviewing Editor and Naama Barkai as the Senior Editor.

Given the number of questions and concerns raised by the three reviewers about the data, their interpretation and conclusions drawn, the manuscript requires significant amount of experimental work before it can be accepted at *eLife*. The reviewers expressed skepticism whether the authors could address them within two months under normal circumstances. Below I summarize the key questions and concerns expressed by the reviewers and the experiments considered by the reviewers as essential. I also include the complete reviews for your perusal.

1. There was a consensus that the proposed model that Nodal signaling activity is not determined by signaling feedback but rather is set by the EGF-CFC co-receptor Oep, to be novel and significant. However, they also thought these main conclusions are quite speculative at this point. A main issue is that the distribution of Nodal ligands in the various experimental conditions is indirectly inferred from the levels of downstream pSmad2 signaling. Specifically, direct observation of the ligands in different oep homozygous and heterozygous mutant and overexpression backgrounds can help to resolve some of the key issues raised by all reviewers. This would rule out that the effects are due to changed balance in positive and negative feedback in signaling rather than ligand distributions. The Schier lab has visualized these ligands in the past (Müller et al., 2012) and could address this issue directly by comparing ligand distributions in WT and MZoep mutants, as well as in oep overexpression condition.

2. The genetic background used for the sensor cells (Mvg1) is a major concern for the analysis and interpretation of these experiments. Although, Vg1 can form heterodimers with both Nodal ligands and is required for their endogenous activity, residual Nodal signaling still persists in MZvg1 mutant embryos (Pellicia et al., *eLife*. 2017). Accordingly, injection of relatively low doses of squint mRNA (10pg) in MZvg1 mutant embryos is sufficient to induce Goosecoid expression (Montague et al., *eLife*. 2017; Pellicia et al., *eLife*. 2017). Therefore, the "sensor" cell assay should be repeated using sqt;cyc double mutant cells to ensure all positive feedback signaling is absent.

3. Moreover, it will be crucial to demonstrate the requirement for ligand-induced acceleration of endocytosis (this is beyond normal endocytosis model that rely on constitutive endocytosis).

4. In addition, there are several simulation questions, which need addressing (please also see the individual reviews below).

a. The rate for λ n is not provided.

b. In supplemental material for the model description: The model requires a very high endocytosis rate for the traveling wave to work.- on the order of 1.7*10^-3^/sec (supplement table). This is 10-20 times faster than estimates for BMP receptors and endocytosis (Pomreinke et al. 2017), however not measured here. This leads to a half-life of 6.8 minutes. That is much faster than other turnover studies from Ed Leof's data for TGF-β endocytosis in a number of cell culture studies inconsistent with modeling studies in TGF-β.

5. The transplanted cells do not seem to remain highly cohesive and rather spread within the blastoderm (Figures 1, 2 and 3). This is an important issue, as a given sensor cell might end up positioned away from the margin, but have initially been located close to the YSL and therefore received a very high dose of Nodal ligands, independently of long-range ligand diffusion. Thus, the authors should analyze sensor cell dispersion post-transplantation and their p-Smad2 signaling status in a more dynamic manner.

6. To support the rationale of Oep loss and traveling wave, the level of maternal RNA over time should be analyzed by RT PCR to quantify maternal RNA loss to see if consistent with timing of wave or cite where this data is available.

7. The authors claim that the p-Smad2 gradient is expanded in MZoep host embryos. However, there are hardly any cells quantified for the control transplants at more than 100µms distance from the margin. The authors should address this issue.

8. Both the number of transplanted sensor cells and their distance to the YSL is highly variable across experimental conditions (compare Figures 1, 2 and 3). This renders the interpretation of the results difficult. Therefore, the authors should perform a detailed analysis of the p-Smad2 behavior as a function of the number of transplanted sensor cells and their distance to the YSL. Perhaps, by binning their data in sub-classes and plotting the variability in p-Smad2 staining across the transplanted cell cluster.

9. Related to that, in the simulations in Figure 4 the size of the clones is very small compared to the experimental data and one cannot get an impression of any position-dependent differences in signaling activity. The level of signaling as a function of position differs in the different experimental conditions – the simulations should allow assessing whether the model adequately captures these changes.

10. A central claim in this manuscript is that the Oep co-receptor critically modulates the diffusion range of the Nodal ligands. The Oep overexpression experiments in Figure 3 show opposite phenotypes. In Figure 3B there is a very large oep overexpressing clone which touches the margin and shows pSmad signaling several cell diameters away from the margin. In Figure 3D a sensor clone in oep overexpressing background shows almost no signal at a comparable distance to the margin. How do the authors reconcile this?

11. The authors state (lines 165-167) that "Loss of Oep led to an expanded range of action of both Cyclops and Squint" based on the "sensor" experiment performed in MZoep;sqt and MZoep;cyc double mutant. However, this conclusion is supported by single images from these double mutants without any quantification. To make this conclusion, such experiments need to be quantified as illustrated in Figure 1B. This is important, as this result would imply that it is Oep that discriminates between Cyc and Sqt ligands and their distinct signaling range, demonstrated by earlier work from the Schier lab and also in this manuscript.

*Reviewer #1:*

1. To address the key question of how the Nodal signaling gradient is formed in zebrafish embryos and the relative contributions of Nodal ligand diffusion and feedback signaling for this process, the authors established a transplantation-based "sensor" cell assay. However, the genetic background used for the sensor cells (Mvg1) is a major concern for the analysis and interpretation of these experiments. Although, Vg1 can form heterodimers with both Nodal ligands and is required for their endogenous activity, residual Nodal signaling still persists in MZvg1 mutant embryos (Pellicia et al., *eLife*. 2017). Accordingly, injection of relatively low doses of squint mRNA (10pg) in MZvg1 mutant embryos is sufficient to induce Goosecoid expression (Montague et al., *eLife*. 2017; Pellicia et al., *eLife*. 2017). Therefore, the "sensor" cell assay should be repeated using sqt;cyc double mutant cells to ensure all positive feedback signaling is abolished.

2. The transplanted cells do not seem to remain highly cohesive and rather spread within the blastoderm (Figures 1, 2 and 3). This is an important issue, as a given sensor cell might end up positioned away from the margin, but have initially been located close to the YSL and therefore received a very high dose of Nodal ligands, independently of long-range ligand diffusion. Thus, the authors should analyze sensor cell dispersion post-transplantation and their p-Smad2 signaling status in a more dynamic manner.

3. Both the number of transplanted sensor cells and their distance to the YSL is highly variable across experimental conditions (compare Figures 1, 2 and 3). This renders the interpretation of the results difficult. Therefore, the authors should perform a detailed analysis of the p-Smad2 behavior as a function of the number of transplanted sensor cells and their distance to the YSL. Perhaps, by binning their data in sub-classes and plotting the variability in p-Smad2 staining across the transplanted cell cluster.

4. A central claim in this manuscript is that the Oep co-receptor critically modulates the diffusion range of the Nodal ligands. In Figures 3c and d, the authors claim that modulating Oep levels in MZsmad2 hosts dramatically reduces the range of p-Smad2 activation in sensor cells. While this seems to be the case when comparing the data in Figures 3c and d, this is not the case when looking at the example for the same control experiment in Figure 1c (Mvg1 → MZsmad2). Why would this be the case?

5. The authors claim that "by facilitating capture of Nodal ligands, Oep regulates range and intensity of the Nodal activity gradient". Although the author's theoretical model is consistent with this interpretation, this remains to be experimentally tested. For this, the authors could analyze the intra- versus extracellular distribution of Nodal ligands ligands in homo- and heterozygous oep mutant embryos, in Zoep mutants and upon oep overexpression in a wt background.

6. Additionally, the authors should experimentally measure the diffusion dynamics of both Nodal ligands in homo- and heterozygous oep mutant embryos, in Zoep mutants and upon oep overexpression in a wt background. Furthermore, it would be interesting to perform similar measurements for Lefty 1 and 2, given the authors suggestion that binding to Oep results in different diffusion ranges for Nodals and Leftys.

7. The dynamics of Oep decay in zygotic oep mutants should be studied in this study.

8. Is the function of Oep in controlling the Nodal signaling range specific? Or would modulating Activin receptor expression produce similar phenotypes?

9. It is unclear why the transplantation assays are reproduced using different parameters from those used to simulate the Nodal signaling gradient in Figure 5. Can the authors comment on why this is? It would be preferable to use similar parameters to reproduce both the transplantation assays and the in vivo gradients.

10. It would be interesting to test whether Lefty overexpression in the MZoep background is sufficient to reduce the Nodal signaling range in sensor cells.

*Reviewer #2:*

In the paper by Lord et al., the authors address two important questions regarding the formation of the Nodal gradient (Cyclops and Squint) from the YSL 6-8 cells into the margin- a pathway that is also regulated by Nodal inhibitors lefty1/2. First, the authors address a recent hypothesis that the nodal signaling gradient is formed by a sequence of positive feedback expression events in a relay that initiates in YSL and that diffusion or transport is not the mechanism of gradient formation. In the process of addressing this, they identify the role for the co-receptor Oep that "sets cell sensitivity" to Nodal. Overall, addressing the feedback question in MZsmad2 embryo hosts with clones of nodal sensitive cells establishes gradient formation in the absence of feedback- this is quite a challenging but convincing experiment to determine the range of Nodal without feedback.

In regards to MZoep mutants, the behavior of MZoep is consistent with co-receptor activity- increasing sensitivity by presumably increasing the formation of receptor competent receptor complexes, while simultaneously impacting gradient spread due to additional ligand capture. Basically- increase the frequency of ligand capture shapes the gradient by reducing the length scale, related to the dimensionless Thiele modulus (ratio of the reaction rate to diffusion rate). The behavior of systems without feedback in the presence of a co-receptor are very well developed and known, so it is unclear what new information is provided by the simulations in figure 4 that behaves similarly to other systems with receptors, binders, and ligand capture. In the model and supplemental materials for the model, the first test assumes "pseudo first-order kinetics"- making the gradient formation a linear ODE at steady-state and it is shown that 4A-it forms a gradient; in 4B- when there's no decay by setting receptor levels to zero, the gradient expands- increasing the rate of decay by receptors reduces the range. Thus 4A-4D really prove things that are widely known in general and have been shown analytically and numerically in many papers. Some papers that have carried out similar but more developed analysis on trapping and/or endocytosis with and without diffusion include the following: (Lander, Nie and Wan, 2002; Lander, 2007) (Coppey et al., 2007; Coppey et al., 2008) (Eldar et al., 2003) (Lander et al., 2020) (Umulis et al., 2006) (Hornung, Berkowitz and Barkai, 2005) some more effort looking at previous co-receptor simulations or mathematical studies that relate binders to gradients should be considered.

Next, in figure 5C, the experiment suggests development of a traveling wave due to receptor depletion. This only occurs in the simulation when there is a "ligand-induced" increase in endocytosis. The preponderance of evidence in cell culture and for other TGF-β ligand-receptor systems, supports an alternative hypothesis- that receptors are constitutively turned over at a constant rate dependent on the turn-over of the membrane via clathrin-mediated endocytosis. The evidence for a positive feedback on the rate of receptor endocytosis in this system that is essential to the underlying hypothesis is not provided. The system that doesn't have ligand-induced endocytosis will not lead to a traveling wave.

The hypothesis for the wave here in the simulation relies on too many unsubstantiated requirements- and is therefore speculative. It requires rapid and ligand-induced endocytosis, loss of receptors and no resupply by maternal transcript (PCR not shown) and operating far from saturation- or it would be a flat signal.

To support the hypothesis, experiments could be attempted to:

1. Determine how close to saturation the receptors are via determining maximum signaling achievable in overexpressed sqt. If it is near saturation, then excess nodal will lead to no increases in PSmad2.

2. Determine the level of maternal RNA over time by RT PCR to quantify maternal RNA loss to see if consistent with timing of wave or cite where this data is available.

3. Block or use endocytosis deficient clones to better determine the validity of the critical ligand-induced endocytosis hypothesis.

4. Identify whether Oep is part of the receptor complex or if it dissociates before signaling- the model is predicated on the co-receptor functioning as the receptor. It is hard to envision a system where to co-receptor is part of the complex for TGF-β and signaling, and then if it is not a part of the complex, then how is ligand-induced rapid endocytosis of the Oep achieved?

In addition there are many simulation questions:

– the rate for λ n is not provided,

In supplemental material for the model description: The model requires a very high endocytosis rate for the traveling wave to work.- on the order of 1.7*10^-3^/sec (supplement table). This is 10-20 times faster than estimates for BMP receptors and endocytosis (Pomreinke et al., 2017), however not measured here. This leads to a half-life of 6.8 minutes. That is much faster than other turnover studies from Ed Leof's data for TGF-β endocytosis in a number of cell culture studies inconsistent with modeling studies in TGF-β.

The units for parameter k1 are not correct in the parameter table- the nodal-receptor association rate. Perhaps, they should be (uM sec)^-1^. Otherwise the units in the differential equations are not consistent. Also the reference is for BMPRII binding rate. The dissociation constant (assuming the rate parameter units is a typo) is then 6.25 X 10^-5^ μm or 6.25*10^-2^ nM. This is quite high and would saturate receptors at 0.12 nM. Here are some constants from Aykul et al.

Ligand Interacting ka(s^-1^M^-1^s^-1^) kd(s^-1^) Kd(nM)

Nodal ACTRIIA 2.0×104 2.0×10-3 100

ACTRIIB ~4.9×104 (est) ~4.9×10-4 (est) ~10 (est)

BMPRII 3.1×105 4.6×10-5 0.149

ALK4 ~4.6×104 (est) ~3.2×10-4 (est) ~15 (est)

ALK7 No Binding

Cripto-1 1.0×104 2.6×10-4 16

Cryptic 5.5×102 1.0×10-3 2,000 †

*Reviewer #3:*

The manuscript by Lord et al., addresses the question of how the Nodal morphogen gradient forms during zebrafish development. This work makes two main points: (1) the Nodal gradient can be established by diffusion alone without a relay of Nodal production; (2) the co-receptor Oep shapes the Nodal gradient and restricts its range by ligand capture. While the presented observations are novel, relevant and important for understanding the mechanism of gradient formation in this system, the conclusions and interpretation require further support. In particular:

1. A main issue is that the distribution of Nodal ligands in the various experimental conditions is indirectly inferred from the levels of downstream pSmad2 signaling. The conclusions are strongly worded, implying proportionality between Nodal ligand levels and pSmad2 levels, eg. in MZSmad2 (line 146-147) and MZoep embryos (line 174-175). However, the pSmad2 levels in this system are dependent not only on the level of activators (Sqt, Cyc), but also on the level of negative regulators such as lefty1 and 2. Although lef1, lef2 are key negative regulators, they might not be the only Nodal-dependent negative feedback on pSmad2. Thus, it is difficult to rule out the possibility that the effects that the authors are seeing on pSmad2 range in the MZSmad2 and MZoep embryos are due to modified negative feedback, rather than distribution of Sqt and Cyc. The Schier lab has visualized these ligands in the past (Müller et al., 2012) and could address this issue directly by comparing ligand distributions in WT and MZoep mutants, as well as in oep overexpression condition.

2. The Oep overexpression experiments in Figure 3 show opposite phenotypes. In Figure 3B there is a very large oep overexpressing clone which touches the margin and shows pSmad signaling several cell diameters away from the margin. In Figure 3D a sensor clone in oep overexpressing background shows almost no signal at a comparable distance to the margin. How do the authors reconcile this?

3. In the simulations in Figure 4 the size of the clones is very small compared to the experimental data and one cannot get an impression of any position-dependent differences in signaling activity. The level of signaling as a function of position differs in the different experimental conditions – the simulations should allow assessing whether the model adequately captures these changes.

4. To draw their conclusions, the authors make certain assumptions about cell rearrangements in this tissue (or the lack thereof). These assumptions are not stated or justified. This needs to be done both for interpreting the results in MZSmad2 and MZoep embryos, but also in the traveling wave experiments in Figure 5 where they look at a later stage (Schield).

[Editors' note: further revisions were suggested prior to acceptance, as described below.]

Thank you for resubmitting your work entitled "The pattern of Nodal morphogen signaling is shaped by co-receptor expression" for further consideration by *eLife*. Your revised article has been evaluated by Naama Barkai (Senior Editor) and a Reviewing Editor.

The manuscript has been improved and the reviewers were satisfied with the revisions. However, the reviewers requested that your response to point 5 (regarding cell rearrangements) should be incorporated into the main text of the manuscript. This is important, so that readers are aware of the assumptions that are made in interpreting the experimental data and know what these assumptions are based on. Not all readers will be familiar with the zebrafish system and this information will make the study more accessible to a wider audience.

---

## [Author Response]

1. There was a consensus that the proposed model that Nodal signaling activity is not determined by signaling feedback but rather is set by the EGF-CFC co-receptor Oep, to be novel and significant. However, they also thought these main conclusions are quite speculative at this point. A main issue is that the distribution of Nodal ligands in the various experimental conditions is indirectly inferred from the levels of downstream pSmad2 signaling. Specifically, direct observation of the ligands in different oep homozygous and heterozygous mutant and overexpression backgrounds can help to resolve some of the key issues raised by all reviewers. This would rule out that the effects are due to changed balance in positive and negative feedback in signaling rather than ligand distributions. The Schier lab has visualized these ligands in the past (Müller et al., 2012) and could address this issue directly by comparing ligand distributions in WT and MZoep mutants, as well as in oep overexpression condition.

We agree that direct visualization of the Nodal ligand gradient would be of great value. However, the reagents required to perform this task for endogenous ligands remain unavailable despite considerable effort. Indeed, it has been a central goal of our and other labs over the past 20 years to visualize endogenous Nodals by generating high-quality 𝛼-Squint and 𝛼-Cyclops antibodies and transgenic lines with fluorescently-tagged ligands. Unfortunately, these efforts have consistently failed due to the presumably very low levels of endogenous Nodal ligands in vivo. The sensor cell assay is the best and currently only answer to these limitations as it represents a very clear and quantitative read-out of Nodal signaling in zebrafish. This assay supports our conclusions about the spatial distribution of Nodal signaling activity.

While we cannot detect endogenous ligands, the reviewers are correct to point out that ligand visualization has been achieved via exogenous overexpression of fluorescently-tagged Nodals^1^. Following the suggestions of the reviewers, we performed two independent experiments to directly visualize fluorescently tagged Nodals in wild-type and MZ*oep* mutants. First, we extended the experiments described in Müller et al., 2012^1^ to visualize ectopic gradients of GFP-tagged Squint produced by transplanted sources in wild-type, Mz*oep* and *oep-*overexpressing MZoep host embryos (Figure 2. Supplement 3). Quantification of these gradients reveals that the Nodal gradient is extended in the absence of Oep. Notably, the gradient was shortened relative to wild type by overexpressing *oep* in MZ*oep* hosts. Second, we used a variant of our sensor cell assay to directly visualize HALOtagged ligands secreted from the YSL (Figure 3 Supplement 1). This approach is akin to the morphotrap approach used in recent studies of morphogen signaling and distribution. In this experiment, mRNAs encoding *vg1-HALO* and squint were injected into the YSL, which corresponds to the endogenous Nodal expression domain. Sensor cells from donor embryos injected with *oep* mRNA were transplanted to the animal pole of these hosts. Ligand captured by the sensor cells was visualized by staining with HALO ligand and subsequent confocal microscopy. Notably, sensor cells accumulate substantial ligand in MZ*oep* hosts (but not wild type), indicating an expanded range of travel from the YSL in the absence of the co-receptor. Taken together, these experiments with fluorescently-tagged, overexpressed ligand corroborate and extend our original findings. We note, however, that the sensor cell assay using pSmad2 as a read-out remains the only approach that enables inference of the distribution of endogenous, untagged ligands.

2. The genetic background used for the sensor cells (Mvg1) is a major concern for the analysis and interpretation of these experiments. Although, Vg1 can form heterodimers with both Nodal ligands and is required for their endogenous activity, residual Nodal signaling still persists in MZvg1 mutant embryos (Pellicia et al., eLife. 2017). Accordingly, injection of relatively low doses of squint mRNA (10pg) in MZvg1 mutant embryos is sufficient to induce Goosecoid expression (Montague et al., eLife. 2017; Pellicia et al., eLife. 2017). Therefore, the "sensor" cell assay should be repeated using sqt;cyc double mutant cells to ensure all positive feedback signaling is absent.

We went to great lengths in using *vg1* mutant sensor cells instead of wild-type cells as sensors to minimize potential relay effects by Vg1-Nodal heterodimers. These heterodimers account for most if not all Nodal signaling in vivo and cannot form in the absence of Vg1. Even if there were very low levels of homodimeric ligands generated, this could not explain the activation of Nodal signaling in sensor cells far from the source, as outlined in detail below.

1. We find no evidence that sensor cells influence one another’s behaviors at a distance. If the M*vg1* sensors were able to propagate signaling via residual positive feedback, we would expect to observe higher signaling levels in sensors with nearby neighbors. However— as detailed in response to comment 8 below— we find that Nodal sensitivity is independent of proximity to other sensor cells. This result further justifies our conclusion that positive feedback is effectively removed in M*vg1* embryos (Figure 2 Supplement 2).

2. We have extensively studied *vg1* mutants in both published^2^ and unpublished work. In M*vg1* embryos, cyclops and squint are expressed at approximately normal levels in the YSL, yet phosphorylated Smad2 is absent or strongly reduced (see updated Figure 1 Supplement 1 in the attached manuscript) and Nodal-induced gene expression is absent^2,3^. These observations show that even at the source of endogenous Nodal signals, potential Nodal homodimers are largely inactive in M*vg1* mutants. It is therefore highly unlikely that M*vg1* sensor cells could propagate signaling via release of Squint or Cyclops homodimers. Moreover, to generate a relay, Nodal-Vg1 heterodimers produced in the YSL of MZ*oep* mutants would still have to travel to the closest sensor cells, which are often located far from the YSL.

3. The reviewers are correct that injection of 10 pg of *squint* mRNA into MZ*vg1* mutant embryos is sufficient to induce *gsc* expression. This argues that Squint can — in principle — be biologically active in the absence of Vg1. However, this activity depends on non-physiological expression levels; a 10 pg mRNA injection is not a ‘relatively low dose’. Calculations using the predicted molecular weight of the in vitro transcribed mRNA indicate that 10 pg corresponds to approximately 12 million transcripts, or about 12,000 transcripts per cell at the 1000-cell stage. We have extensively quantified squint transcript abundance using single-molecule FISH in wild-type embryos. In the blastula, the highest expressing cells contain only ~50 molecules. We note that Pellicia *et al.* found reducing the dose to 1 pg of *squint* mRNA (~1,200 transcripts per cell at the 1000-cell stage) —though still a large amount — was insufficient to induce Nodal target expression in *vg1* mutants^3^. It is therefore virtually impossible for sensor cells—even if they are subject to high levels of Nodal signaling— to express squint in the amounts required to drive homodimeric Nodal signaling.

3. Moreover, it will be crucial to demonstrate the requirement for ligand-induced acceleration of endocytosis (this is beyond normal endocytosis model that rely on constitutive endocytosis).

The reviewers are correct that our model used ligand-dependent increase in the rate of endocytosis as a proxy for receptor clearance. We are happy to include alternative models because this particular molecular implementation is not strictly required to reproduce our major findings (e.g. wave formation in Z*oep* mutants). Rather, the key requirement is that ligand binding increases the effective rate of clearance of the receptor. We did not intend to make strong mechanistic claims about Oep endocytosis, trafficking or degradation. Our major finding is that Oep has unappreciated functions— regulation of ligand range through capture— that make the regulation of its expression particularly interesting. This led us to the striking discovery of the Z*oep* signaling wave. The purpose of the original model was to build intuition for this phenomenon using the simplest principles available.

Following the reviewers’ suggestions, we formulated a new model in which bound and unbound receptors are internalized with equal rates, but the rate of lysosomal trafficking is increased by ligand occupancy. In the newly-added Figure 5 Supplement 1 we show that this trafficking-oriented model also predicts a signaling wave in the Z*oep* mutants. We note that mechanistic basis exists to include ligand-dependent trafficking rates in the model. Activation-dependent receptor downregulation is a well recognized feature of TGF-b signaling^4-6^ and is included in published quantitative models of its function^7^. Importantly, activity-enhanced degradation of Nodal receptors has been described in zebrafish for the stages we examine in the paper^8^.

We have added the trafficking-oriented model to the resubmitted manuscript as a supplementary figure. We have also revised the manuscript to include a discussion of the molecular mechanisms that can support wave formation (Supplemental Text: Gradient simulations accounting for receptor trafficking).

4. In addition, there are several simulation questions, which need addressing (please also see the individual reviews below).

We thank the reviewers for their thoughtful evaluation of our model. We address the specific concerns highlighted by the editor below.

a. The rate for λ n is not provided

We apologize for the oversight. We selected a Nodal synthesis rate of 8 x 10^-6^ µM/s at the source. This corresponds to an approximate synthesis rate of ~3 molecules/min per cell within the source domain.

b. In supplemental material for the model description: The model requires a very high endocytosis rate for the traveling wave to work.- on the order of 1.7*10^-3^/sec (supplement table). This is 10-20 times faster than estimates for BMP receptors and endocytosis (Pomreinke et al. 2017), however not measured here. This leads to a half-life of 6.8 minutes. That is much faster than other turnover studies from Ed Leof's data for TGF-β endocytosis in a number of cell culture studies inconsistent with modeling studies in TGF-β.

There are several studies that support the plausibility of our endocytosis rate. We are aware of the TGF-b system measurements from the Leof lab, however, embryonic Nodal signaling differs from TGF-b signaling in differentiated cells in many respects. We therefore based our parameter choice on measurements from the Gurdon lab that were carried out in *Xenopus* animal cap cells using Activin, a signal from the Nodal family of TGF-b ligands^9^. In this more relevant context, antibody uptake experiments demonstrate that labeled surface receptors are completely internalized within 30 minutes. Rapid endocytosis of Nodal receptors is also supported by measurements in cell culture using a *bona fide* Nodal ligand^10^; pulse-chase experiments demonstrate that bound Nodal ligands are completely internalized and degraded within 60 minutes of presentation. We believe that these studies are sufficient to motivate our choice of internalization rate.

Further, the suggested rate for BMP receptor internalization is not strongly supported. The cited paper^11^ does not contain any measurements of BMP receptor stability. The value the reviewer appears to be referring to (λBMP = 8.9 × 10^−5^/s, approximately 20-fold lower than the endocytosis rate in our original model) represents the measured stability of overexpressed BMP2b-Dendra2 in an FDAP experiment. Given the availability of high-quality, direct measurements of receptor endocytosis rates for Nodal family ligands—both in embryos and cell culture— we do not see the value in relying on indirect measurements derived for a different ligand family.

That said, wave formation in Z*oep* mutants can be accommodated using slower rates of receptor internalization as suggested by the reviewer. The trafficking-oriented model (discussed in comment 3 above) assumes a constitutive receptor internalization rate of 2.7 x10^-4^/s, giving an average lifetime of ~1 hour on the cell surface for both bound and unbound receptors. This parameter selection still leads to a stable signaling gradient in ‘wild type’ simulations and a traveling wave in ‘Z*oep*’ simulations (see Figure 5 Supplement 1 in the attached revised manuscript). The key predictions of our models are therefore robust to substantial variation in internalization rate.

We have added an explanation for this parameter choice to the supplementary text (Supplemental Text: Gradient simulations accounting for receptor trafficking).

5. The transplanted cells do not seem to remain highly cohesive and rather spread within the blastoderm (Figures 1, 2 and 3). This is an important issue, as a given sensor cell might end up positioned away from the margin, but have initially been located close to the YSL and therefore received a very high dose of Nodal ligands, independently of long-range ligand diffusion. Thus, the authors should analyze sensor cell dispersion post-transplantation and their p-Smad2 signaling status in a more dynamic manner.

We shared the reviewers concerns about the role of cell movement in the sensor cell assay. This consideration was the purpose of performing control transplants into wild-type embryos. If transplanted sensors were accumulating Nodal close to the margin before moving toward the animal pole, we would observe high pSmad2 staining sensor cells far from the margin in wild-type embryos. This was not the case (see e.g. manuscript Figure 1B). Further, sensors transplanted directly to the animal pole of MZ*oep* embryos still detected Nodal (Figures 2D,E). Explaining this result with migration would require the cells to traverse the entire embryo twice—down to the margin and back again— while retaining a memory of their Nodal signaling history. These experiments suggest that migration plays at best a very minor role in the readout of the sensor cells.

Additionally, our lab has performed extensive analysis of cell movement over the past decade. Endogenous marginal cells maintain their position relative to the margin during blastula stages^12^. Cells transplanted to the margin at early blastula stages— into wild-type or MZ*oep* hosts— only begin to migrate upward after internalization during gastrulation^13^. Given that our analysis is carried out in blastula stages (i.e. prior to internalization), cell migration is a negligible concern.

6. To support the rationale of Oep loss and traveling wave, the level of maternal RNA over time should be analyzed by RT PCR to quantify maternal RNA loss to see if consistent with timing of wave or cite where this data is available.

We thank the reviewers for the comment and regret omission of the relevant citation in the original manuscript. Zhang et al. showed that maternal oep mRNA is no longer detectable at the germ ring stage in zygotic *oep* mutants^14^ (see Zhang et al., Cell 1998; Figure 5L). Moreover, a recent paper from the Vastenhouw lab^15^ measured *oep* transcript stability in the absence of zygotic transcription. They report that maternal transcripts are undetectable by 60% epiboly (within ~30 minutes of the final time point included in Figure 5). Further, they noted over-representation of ‘destabilizing’ codons in the *oep* transcript, consistent with rapid turnover. These findings suggest that the maternal *oep* supply is exhausted in the correct time window to support wave formation. We have added these considerations to the revised manuscript.

7. The authors claim that the p-Smad2 gradient is expanded in MZoep host embryos. However, there are hardly any cells quantified for the control transplants at more than 100µms distance from the margin. The authors should address this issue.

We agree that the paper would be improved by a more formal treatment of this issue. A statistical analysis indicates that the differences between the two gradients are significant. The difference in average staining intensity of sensor cells falling between 100 and 150 µm from the margin in wild-type and MZ*oep* backgrounds is significant (Welch’s t-test, *p* = 4.4e-11). The difference between all cells in wild-type hosts (both host-derived and donor-derived) and sensor cells in the MZ*oep* background within this spatial domain also tests as significant (Welch’s t-test, *p* = 2.2e-12). Further, comparing fit parameters for single-exponential decay models supports our conclusion that the gradient is significantly expanded in MZ*oep* mutants (exponential decay constants +95% confidence bounds are -0.02 + 0.004 µm^-1^ and -0.007 + 0.002 µm^-1^ for wildtype and MZ*oep*, respectively). We therefore do not find it likely that the differences between wild-type and MZ*oep* embryos arises due to sampling issues associated with the transplant assay. We have added these considerations to the revised manuscript.

8. Both the number of transplanted sensor cells and their distance to the YSL is highly variable across experimental conditions (compare Figures 1, 2 and 3). This renders the interpretation of the results difficult. Therefore, the authors should perform a detailed analysis of the p-Smad2 behavior as a function of the number of transplanted sensor cells and their distance to the YSL. Perhaps, by binning their data in sub-classes and plotting the variability in p-Smad2 staining across the transplanted cell cluster.

We agree with the reviewers that this is a potentially interesting dimension to our dataset. We have performed the analysis and added it to the revised manuscript as Figure 2 Supplement 2.

In short, cell clustering does not appear to alter the behavior of the M*vg1* sensors. We calculated the deviation of each sensor’s pSmad2 staining intensity from the average intensity given its position (i.e. the residual with respect to the red curve in Figure 2b). If sensor cell clustering led to an increase in Nodal sensitivity—for example, through positive feedback interactions— we would expect to see cells with more neighbors have higher residuals. However, we observe no clear relationship between these quantities (Figure 2 Supplement 2C). Further, we find no relationship between the total number of sensor cells transplanted into each embryo and their signaling behavior (Figure 2 Supplement 2D). These results support the use of the sensor cell assay as a reliable readout of the Nodal gradient.

9. Related to that, in the simulations in Figure 4 the size of the clones is very small compared to the experimental data and one cannot get an impression of any position-dependent differences in signaling activity. The level of signaling as a function of position differs in the different experimental conditions – the simulations should allow assessing whether the model adequately captures these changes.

We thank the reviewer for the comment. To make the figures more easily comparable with the experimental data, we performed new simulations in which an increased number of sensors are randomly distributed throughout the field. By quantifying the signaling intensity in these sensors across replicate simulations, we compiled ‘gradient’ measurements comparable to the plots in Figures 1-3. These simulations illustrate the key features of the experimental data—i.e. *oep* absence extends the gradient, while *oep* overexpression shortens it— that are required to understand the model presented in Figure 5. The updated analyses have been added to Figure 4 in the revised main text.

10. A central claim in this manuscript is that the Oep co-receptor critically modulates the diffusion range of the Nodal ligands. The Oep overexpression experiments in Figure 3 show opposite phenotypes. In Figure 3B there is a very large oep overexpressing clone which touches the margin and shows pSmad signaling several cell diameters away from the margin. In Figure 3D a sensor clone in oep overexpressing background shows almost no signal at a comparable distance to the margin. How do the authors reconcile this?

This concern appears to arise from a misunderstanding. Both experiments are consistent with Oep regulating the diffusion range of Nodal ligands. In Figure 3B, the sensor cells overexpress Oep. This results in enhanced ligand capture and hypersensitivity to Nodal in the sensors. In Figure 3D the host cells overexpress Oep. As the host cells—which lack Smad2— capture Nodal at an enhanced rate, less ligand is available for the sensor cells, resulting in little detectable signaling activity. These phenotypes are two sides of the same coin and are the purpose of the figure. We also note that both of these effects are reproduced in the computational model of Figure 4 (see panels C and D in the attached revised manuscript). We have clarified these conclusions in the revised manuscript.

11. The authors state (lines 165-167) that "Loss of Oep led to an expanded range of action of both Cyclops and Squint" based on the "sensor" experiment performed in MZoep;sqt and MZoep;cyc double mutant. However, this conclusion is supported by single images from these double mutants without any quantification. To make this conclusion, such experiments need to be quantified as illustrated in Figure 1B. This is important, as this result would imply that it is Oep that discriminates between Cyc and Sqt ligands and their distinct signaling range, demonstrated by earlier work from the Schier lab and also in this manuscript.

The revised manuscript now contains gradient quantifications for MZ*oep;sqt* and MZ*oep;cyc* mutant embryos (see revised Figure 2 Supplement 1). These quantifications bear out that both Squint and Cyclops signal over a long range in the absence of Oep; the sensor cells exhibit clearly increased pSmad2 staining over background even 150 µm from the YSL. However, we stress that we do not claim that Oep has differential effects on Squint and Cyclops, nor do we believe that this point is crucial to the paper. Indeed, we are currently agnostic as to whether Oep is responsible for the range difference between Squint and Cyclops. The purpose of this figure was to demonstrate that both Cyclops and Squint can signal over a long range in the absence of Oep.

References:

1. Muller, P. *et al.* Differential diffusivity of Nodal and Lefty underlies a reaction-diffusion patterning system. *Science* 336, 721-724, doi:10.1126/science.1221920 (2012).

2. Montague, T. G. & Schier, A. F. Vg1-Nodal heterodimers are the endogenous inducers of mesendoderm. *Elife* 6, doi:10.7554/eLife.28183 (2017).

3. Pelliccia, J. L., Jindal, G. A. & Burdine, R. D. Gdf3 is required for robust Nodal signaling during germ layer formation and left-right patterning. *Elife* 6, doi:10.7554/eLife.28635 (2017).

4. Anders, R. A., Arline, S. L., Dore, J. J. & Leof, E. B. Distinct endocytic responses of heteromeric and homomeric transforming growth factor beta receptors. *Mol Biol Cell* 8, 2133-2143, doi:10.1091/mbc.8.11.2133 (1997).

5. Mitchell, H., Choudhury, A., Pagano, R. E. & Leof, E. B. Ligand-dependent and independent transforming growth factor-beta receptor recycling regulated by clathrinmediated endocytosis and Rab11. *Mol Biol Cell* 15, 4166-4178, doi:10.1091/mbc.e04-030245 (2004).

6. Dore, J. J., Jr. *et al.* Mechanisms of transforming growth factor-beta receptor endocytosis and intracellular sorting differ between fibroblasts and epithelial cells. *Mol Biol Cell* 12, 675-684, doi:10.1091/mbc.12.3.675 (2001).

7. Vizan, P. *et al.* Controlling long-term signaling: receptor dynamics determine attenuation and refractory behavior of the TGF-beta pathway. *Sci Signal* 6, ra106, doi:10.1126/scisignal.2004416 (2013).

8. Zhang, L. *et al.* Zebrafish Dpr2 inhibits mesoderm induction by promoting degradation of nodal receptors. *Science* 306, 114-117, doi:10.1126/science.1100569 (2004).

9. Jullien, J. & Gurdon, J. Morphogen gradient interpretation by a regulated trafficking step during ligand-receptor transduction. *Genes Dev* 19, 2682-2694, doi:10.1101/gad.341605 (2005).

10. Le Good, J. A. *et al.* Nodal stability determines signaling range. *Curr Biol* 15, 31-36, doi:10.1016/j.cub.2004.12.062 (2005).

11. Pomreinke, A. P. *et al.* Dynamics of BMP signaling and distribution during zebrafish dorsal-ventral patterning. *Elife* 6, doi:10.7554/eLife.25861 (2017).

12. Dubrulle, J. *et al.* Response to Nodal morphogen gradient is determined by the kinetics of target gene induction. *Elife* 4, doi:10.7554/eLife.05042 (2015).

13. Carmany-Rampey, A. & Schier, A. F. Single-cell internalization during zebrafish gastrulation. *Curr Biol* 11, 1261-1265, doi:10.1016/s0960-9822(01)00353-0 (2001).

14. Zhang, J., Talbot, W. S. & Schier, A. F. Positional cloning identifies zebrafish one-eyed pinhead as a permissive EGF-related ligand required during gastrulation. *Cell* 92, 241251, doi:10.1016/s0092-8674(00)80918-6 (1998).

15. Vopalensky, P., Pralow, S. & Vastenhouw, N. L. Reduced expression of the Nodal coreceptor Oep causes loss of mesendodermal competence in zebrafish. *Development* 145, doi:10.1242/dev.158832 (2018).

[Editors' note: further revisions were suggested prior to acceptance, as described below.]

The manuscript has been improved and the reviewers were satisfied with the revisions. However, the reviewers requested that your response to point 5 (regarding cell rearrangements) should be incorporated into the main text of the manuscript. This is important, so that readers are aware of the assumptions that are made in interpreting the experimental data and know what these assumptions are based on. Not all readers will be familiar with the zebrafish system and this information will make the study more accessible to a wider audience.

We are grateful to the reviewers for their thorough review and we are happy to hear that they were satisfied with our revisions. We agree that the manuscript would be improved by adding a discussion of cell rearrangements in the sensor cell assay to the main text. We have made the following changes:

1. Cell rearrangement in the sensor cell assay is now explicitly addressed in the first Results section. We describe the controls used to rule out problems caused by cell rearrangements after transplant. Additionally, we added citations from the literature to support our claim that rearrangement is minimal prior to gastrulation in zebrafish.

2. We clarified and expanded the paragraph on relay-mediated spread of Nodal signaling in the discussion to incorporate insights from a recent preprint.

3. We expanded the paragraph on signaling waves in the discussion to provide better context for our discovery of a depletion wave in Z*oep* mutants.